# Bioinspired structural hydrogels with highly ordered hierarchical orientations by flow-induced alignment of nanofibrils

Shuihong Zhu ®[1,2], Sen Wang[1], Yifan Huang[3], Qiyun Tang[3], Tianqi Fu[1], Riyan Su[4], Chaoyu Fan[1], Shuang Xia[1], Pooi See Lee ®[2] ✉ & Youhui Lin ®[1,5] ✉

Natural structural materials often possess unique combinations of strength and toughness resulting from their complex hierarchical assembly across multiple length scales. However, engineering such well-ordered structures in synthetic materials via a universal and scalable manner still poses a grand challenge. Herein, a simple yet versatile approach is proposed to design hierarchically structured hydrogels by flow-induced alignment of nanofibrils, without high time/energy consumption or cumbersome postprocessing. Highly aligned fibrous configuration and structural densification are successfully achieved in anisotropic hydrogels under ambient conditions, resulting in desired mechanical properties and damage-tolerant architectures, for example, strength of $14 \pm 1$ MPa, toughness of $154 \pm 13$ MJ m$^{-3}$, and fracture energy of $153 \pm 8$ kJ m$^{-2}$. Moreover, a hydrogel mesoporous framework can deliver ultra-fast and unidirectional water transport (maximum speed at 65.75 mm s$^{-1}$), highlighting its potential for water purification. This scalable fabrication explores a promising strategy for developing bioinspired structural hydrogels, facilitating their practical applications in biomedical and engineering fields.

The design and fabrication of hydrogels have been widely explored in soft electronics, tissue engineering, and implantable devices, which is attributed to the similarity between synthetic hydrogels and biological systems[1–5]. Conventional synthetic hydrogels typically maintain isotropic structures with randomly oriented polymer networks, because their preparation typically involves dissolving the polymer precursor in an aqueous solution. To some extent, the presence of disordered building blocks in synthetic composites can facilitate sensing and actuation[6,7]. However, in biological systems, numerous natural structural materials (such as wood, muscle, tendon, skeleton, and shell) exhibit hierarchically ordered structures that play a crucial role in adaptation to complex environments, including mass transport, actuation, self-defense, and surface lubrication[8–11]. Mimicking such

well-ordered hierarchical structures in synthetic hydrogels presents a significant challenge since the polymer chains are homogeneously dissolved in aqueous environments[12,13].

The integration of molecular and structural engineering approaches has been adopted to design biomimetic anisotropic hydrogels with sophisticated ordered structures similar to their biological counterparts, including electric/magnetic-field orientation[14], compositing strategy[15–18], freeze-casting[19–22], strain alignment[23–25], and self-assembly[26–28]. For instance, by introducing natural well-organized hierarchical architectures (such as wood) or alien fiber reinforcements into hydrogel networks, anisotropic composite hydrogels can be developed[16,18]. Despite these nanocomposite hydrogels having high tensile strength compared to homogeneous tough hydrogels, their

[1]Department of Physics, Research Institute for Biomimetics and Soft Matter, Fujian Provincial Key Laboratory for Soft Functional Materials Research, Xiamen University, Xiamen 361005, PR China. [2]School of Materials Science and Engineering, Nanyang Technological University, 50 Nanyang Avenue, Singapore 639798, Singapore. [3]Key Laboratory of Quantum Materials and Devices of Ministry of Education, School of Physics, Southeast University, Nanjing 211189, PR China. [4]Shandong Huankeyuan Environmental Testing Co., Ltd, Jinan 250013, PR China. [5]National Institute for Data Science in Health and Medicine, Xiamen University, Xiamen 361102, PR China. ✉e-mail: pslee@ntu.edu.sg; linyouhui@xmu.edu.cn

stretchability and water content are limited, which greatly impedes their application in engineering hydrogels. The ice-templating method is broadly employed to create anisotropic structures in hydrogels because of its versatility for various polymers, but it is often associated with inferior mechanical performance when used alone[22,29,30]. Combining ice templating with post-treatments, such as annealing and salting out, can effectively strengthen and toughen hydrogels[20,31]. These approaches can improve the mechanical performance of hydrogels over a wide range, yet they suffer from time/energy consumption during freezing or other processing, as well as insufficient molecular-scale organization. Besides, strain alignment has been widely regarded as a kind of ideal method to fabricate anisotropic hydrogels. Nanofibril alignment induced by mechanical strain, such as drying in confined condition and mechanical training, have been proposed to generate anisotropic hydrogels with perfectly aligned fibrous structures that resemble biological tissues[23,24]. Nevertheless, these structural engineering methods require tedious and repeated drying processes and cyclic stretching, which greatly hinder their industrial-scale production. Until now, a facile and universal approach for developing biomimetic hydrogels with hierarchical architecture at multiple length scales under ambient conditions remains an open issue. Alternatively, flow-induced alignment can serve as an efficient method to regulate nanocomposites at the molecular scale, holding prospective insights to design high-performance structural hydrogels in a scalable manner[32–34].

Here, we present a scalable strategy to fabricate bioinspired structural hydrogels with highly ordered hierarchical orientations via flow-induced alignment of nanofibrils. Due to its long-range ordered and highly dense structures, anisotropic fibrous hydrogel (AFH) exhibits remarkable mechanical properties and damage-tolerant architectures compared to its isotropic counterparts. Concretely, anisotropic fibrous hydrogels simultaneously achieve a tensile strength of up to $14 \pm 1$ MPa, a toughness of $154 \pm 13$ MJ m$^{-3}$, and a fracture energy of $153 \pm 8$ kJ m$^{-2}$. Moreover, the unique microchannel structure in AFH opens up possibilities for the development of ultrafast and anisotropic mass transport (maximum speed at 65.75 mm s$^{-1}$), enabling efficient harvesting of clean water. This flow-induced alignment can serve as a time- and energy-saving strategy for fabricating bioinspired structural hydrogels within molecular-scale precision, elucidating its capacity for fast and large-scale production. This work represents a significant advancement in the development of hierarchically structured materials and opens up avenues for further exploration and innovation in this captivating research area.

## Results

### Material design and characterization

Owing to its excellent hydrophilicity and programmable engineering, poly(vinyl alcohol) (PVA) was employed as the exemplary hydrogel material. We chose ammonium sulfate solution as the coagulation since it possesses low volatility, high water solubility, and is cost-effective and environmentally friendly compared to other non-solvent coagulants. The random PVA polymer chains were first subjected to flow-induced alignment and then fostered strong aggregation and crystallization by a kosmotropic salt solution, leading to an anisotropic single-composition hydrogel with a highly compact and well-aligned structure (Fig. 1a). Figure 1b illustrates the multiscale hierarchical structures of the AFH spanning from the molecular to the macroscopic scale. At the microscale, the anisotropic fibrous hydrogel was composed of hydrogel fibers with a tightly stacked alignment (Fig. 1c). As a comparison, isotropic PVA hydrogels prepared by freezing-thawing (FT) and freezing-soaking (FS) were adopted as control samples. The randomly distributed structures of FT hydrogel and FS hydrogel were confirmed through scanning electron microscope (SEM) images (Supplementary Fig. 1). As shown in Fig. 1d, e, nanoscale aggregated polymer networks exhibited aligned mesh-like nanofibril network

along the injection direction due to the strong self-aggregation and phase separation of PVA chains in the coagulation. Analogous anisotropic ordered structures generally play crucial roles in biological systems due to their excellent mechanical properties and unique functions. The corresponding macroscopic views of integrated hydrogel fibers and single fiber were shown in Fig. 1f and g, respectively, indicating that the anisotropic fibrous hydrogel maintained a robust structure. Confocal microscopy and atomic force microscope (AFM) of the anisotropic fibrous hydrogel visually demonstrated a well-developed alignment resembling its natural counterparts (Fig. 1h, Supplementary Figs. 2 and 3).

### Preparation of bioinspired structural hydrogels

A consecutive fabrication process to develop the hierarchical assembly of anisotropic structures was depicted schematically in Fig. 2a. This flow-induced alignment was inspired by the wet-spinning process, a well-established method for continuous fiber fabrication in the textile industry. Specifically, a PVA aqueous solution was injected into 3 M ammonium sulfate and stretched to form hydrogel fibers. According to the theoretical research on wet spinning[33,35,36], the shear flow controls the orientation of the polymer chains within the spinneret, where a velocity gradient perpendicular to the fluid velocity is distributed. This gradient of the velocity field creates shear forces, which tend to align the polymer chains along the direction of flow; that is shear-flow-induced alignment of nanofibrils. In addition to the shear flow, extensional flow in the coagulation bath plays a crucial role in determining the orientation and properties of polymers, which refers to a velocity gradient along the fluid velocity. During extensional flow, the polymer chains are stretched and aligned parallel to the flow direction, which results in an increase in the degree of molecular orientation in the composites; this is stretch-induced orientation. As a result, fiber shaping and orientation were dominated by both internal and external induction and alignments.

To illustrate the influence of these two flow fields on the orientation of PVA chains in this work, computer simulations were performed. We simulated the PVA polymer chain using a coarse-grained model. The orientation of the PVA chain was quantified using the end-to-end distance, with the x-axis representing the shear or elongation direction. The shear flow and extensional flow were analyzed separately to investigate the effects of these two flow fields on the alignment of PVA chains (details in Supplementary Movie 1 and Supplementary Methods). At low polymer concentration and shear rate, with a concentration of 10 wt% PVA aqueous solution and an injection speed of 0.6 ml min$^{-1}$, the PVA chains experienced minimal deformation along the x-axis due to shear flow. As the polymer concentrations and shear rates increased, reaching a concentration of 25 wt% for the PVA aqueous solution and an injection speed of 1.75 ml min$^{-1}$, the orientation of PVA chains induced by shear flow became evident (Fig. 2b, d). Moreover, we varied the volume ratios of PVA chains to investigate the impact of extensional flow on the orientation of PVA chains. Accordingly, the extensional flow in the coagulation bath was responsible for orienting the polymer, and as stretching proceeded, the polymer chains became more aligned (Fig. 2c, e). Based on the results of molecular dynamical simulations and the spinning parameters in this work, it has been found that the orientation of the PVA chains during the spinning process is primarily influenced by the extensional flow (Supplementary Fig. 4).

To study the impact of spinning parameters on the structure of hydrogel fibers, we employed various spinneret diameters and draw ratios to achieve flow-induced orientation. As depicted in the SEM images (Supplementary Fig. 5), the fiber diameter gradually decreases with increasing draw ratio and decreasing nozzle diameter. Notably, the conventional wet-spinning process usually includes several post-cleaning steps to prevent the fibers from sticking together, whereas

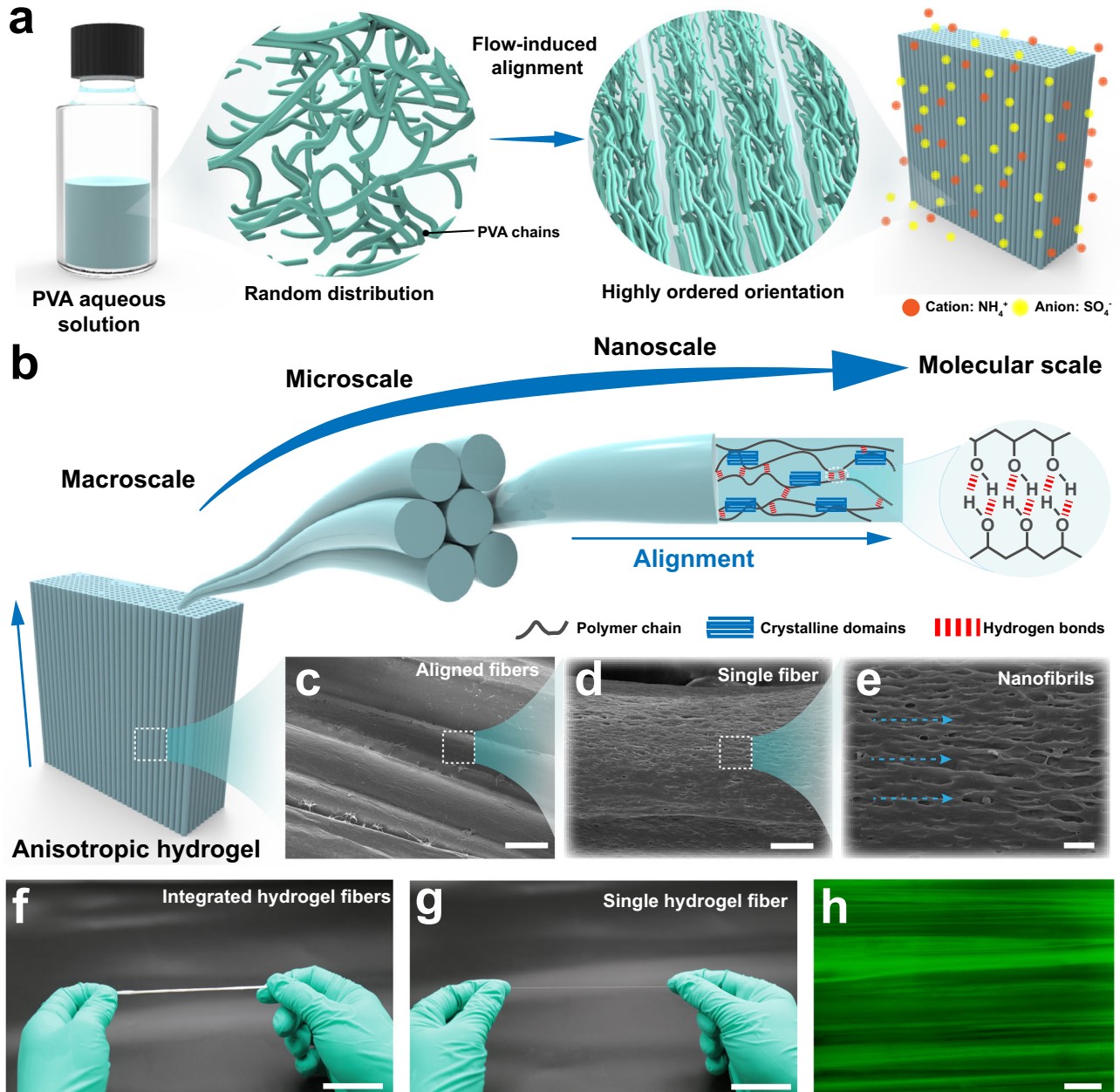

**Fig. 1 | Highly aligned hierarchical fibrous hydrogels inspired by natural structural materials. a** Randomly distributed PVA polymer chains transform into a highly oriented hydrogel network by flow-induced alignment. **b** Schematic illustration of anisotropic hydrogels with multiscale hierarchical architectures from macroscale to molecular scale. Multiscale magnified SEM images showing the preferentially aligned fiber structures. The scale bars for **c**–**e** are 100, 10, and 2 μm, respectively. **f**, **g** Macroscopic view of integrated hydrogel fibers and single hydrogel fiber. Scale bar = 2 cm. **h** Confocal images showing the aligned structures of anisotropic hydrogels. Scale bar = 150 μm.

the strategy we proposed here relies on the collection and assembly of the hydrogel fibers directly after a short-distance coagulation bath (Supplementary Fig. 6). The reeling bobbin driven by a servo motor was employed to tightly assemble the spun hydrogel fibers, where a certain number of hydrogen bonds might form between adjacent nascent fibers (Fig. 2e). This is attributed to the fact that the short-distance coagulation has not fully shaped the hydrogel fibers through the salting-out effect, resulting in the assembly of adjacent as-spun fibers, which can be verified by microscopic morphology and mechanical characterization. Subsequently, the resultant hydrogels with highly aligned fibrous structures were cut and immersed in a kosmotropic salt solution to achieve structural densification via hydrogen bonds and crystalline domains (Supplementary Fig. 7). The

water content of the samples was variable during the salting-out process (Supplementary Fig. 8).

An anisotropic, flexible, and stretchable hydrogel could be deformed arbitrarily without apparent change to its structure (Fig. 2f and Supplementary Fig. 7). The SEM images in Fig. 1e revealed that the nanostructured PVA polymer chains were preferentially aligned with the longitudinal direction (L direction) of hydrogel fibers. In addition, the obtained hydrogel fibers displayed an intriguing sheath-core structure that is typically observed in conventional wet-spun fibers and some natural fibers, such as spider silk[37,38]. The R-view of the AFH was shown in Fig. 2g, which was the cross-section cut along the radial direction (R direction) of hydrogel fibers, indicating the dense assembly of the spun hydrogel fibers with microchannels. The SEM

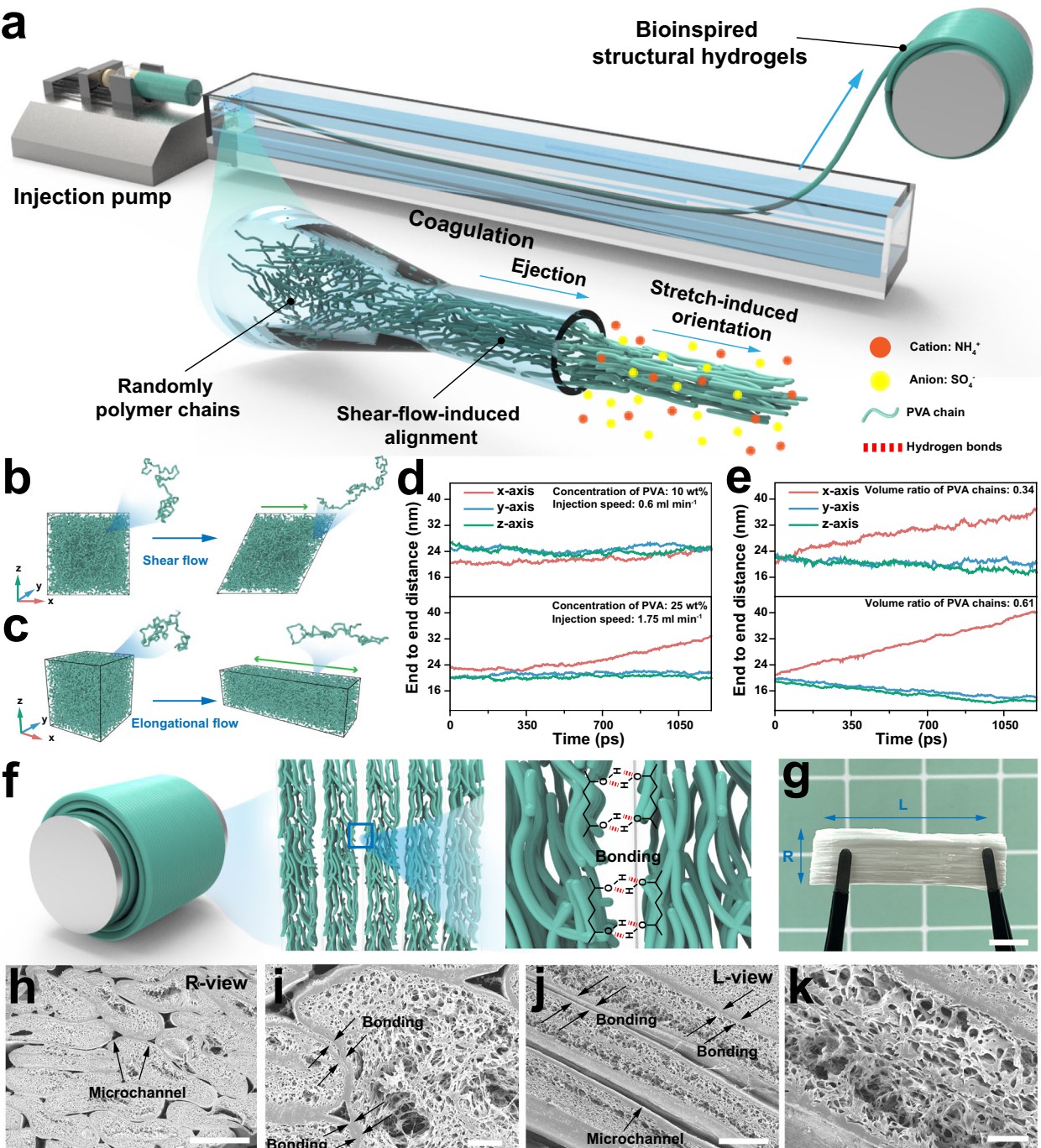

**Fig. 2 | Design strategy for fabricating bioinspired structural hydrogels.**
**a** Fabrication procedure for well-ordered nanofibrils using flow-induced alignment. Molecular dynamics simulations of PVA chains in **b** the injection process and **c** stretching process. The Cartesian coordinate components of the end-to-end distance of PVA chains under **d** two shear flow conditions and **e** extensional flow at various volume ratios. **f** Schematic diagram of the bonding between hydrogel fibers. **g** Typical photographs of anisotropic fibrous hydrogels. Scale bar = 1 cm. **h** SEM cross-sectional image of anisotropic fibrous hydrogels in the R direction. Scale bar = 300 μm. **i** Magnified SEM image of **h**. Scale bar = 40 μm. **j** SEM cross-sectional image of anisotropic fibrous hydrogels in the L direction. Scale bar = 100 μm. **k** Magnified SEM image of **j**. Scale bar = 20 μm.

image in Fig. 2h revealed a sheath-core structure with a dense outer layer and a sparse core, which might be attributed to the different tensions impacted on the polymer chains outside and inside the hydrogel fibers during the spinning process, or the uneven distribution of the coagulant in the hydrogel fibers stemmed from short-distance coagulation[39]. The cross-section cut along the L direction (L-view) was also presented to verify the bonding between hydrogel fibers, as well as the biomimetic sheath-core structure (Fig. 2i, j). This

unique structure of AFH bears a slight resemblance to the innate microchannels found in plant stems, holding great potential as an anisotropic mesoporous microfluidic framework for achieving unidirectional mass transport.

Since wet spinning technology has been highly industrialized, this flow-induced alignment strategy is expected to enable the mass production of structural materials. Spinnerets with varying numbers of nozzles were utilized to perform the spinning process and assess the

potential for large-scale production of biomimetic hydrogels (Supplementary Fig. 9). A nozzle with a 0.65 mm inner diameter took approximately 30 min to complete the spinning of 20 ml PVA aqueous solution at an injection speed of 0.6 ml min$^{-1}$. As the number of nozzles and injection speed increased, the efficiency of AFH preparation significantly improved (details in Supplementary Table 1). In this work, we utilized a limited number of nozzles to demonstrate the applicability of wet spinning in preparing anisotropic hydrogels. In fact, industrial wet spinning usually uses spinnerets with thousands of nozzles, which can be customized according to the specific practical application, making large-scale production of anisotropic fibrous hydrogels feasible.

## Characterization of nanofibrillar structures

To investigate nanofibrillar structures, wide-angle X-ray scattering (WAXS) was used to reveal the crystalline domains of the resulting hydrogels (Fig. 3a–c). According to the WAXS results, no noticeable alignment of crystalline domains was observed in the FT hydrogel and FS hydrogel. On the other hand, the crystalline domains in AFH exhibited a highly oriented microstructure (Fig. 3d). As shown in Fig. 3e, the AFH displayed a steep diffraction peak at $2\theta = 19.8°$, which corresponds to the typical reflection plane of $(10\bar{1})$ in semicrystalline PVA[40–42]. Besides, small peaks at $2\theta = 16.5°$, $18.8°$, and $22.1°$ are also observed in the hydrogel prepared by salting out, depicting a higher crystallinity in AFH compared to FT hydrogels and FS hydrogels. To further quantify the structural characteristics of AFH, differential scanning calorimetry (DSC) was performed to measure the crystallinities of the resulting hydrogel after 2 h and 24 h salting out (Fig. 3f and Supplementary Fig. 10). Excess chemical cross-links were introduced to prevent further crystallization prior to the DSC measurement. As a result, the FT hydrogels showed negligible endothermic peaks, with a low crystallinity of 0.56 % in the dry state, and FS hydrogels displayed a crystallinity of 6.92 % in the dry state (Fig. 3g). In contrast, AFH possessed elevated crystallinities of 21.09% in a dry state and 8.94% in a wet state. This may be attributed to its unique fibrous structure, which can facilitate ion diffusion in hydrogel networks. This trend was consistent with the WAXS results, which supported the structural densification of the hydrogel networks.

Moreover, in situ small-angle X-ray scattering (SAXS) was utilized to further analyze the crystalline morphology of AFH during stretching. As shown in Fig. 3h and Supplementary Fig. 11, no distinct peak appeared in the SAXS profiles of FT hydrogels, indicating negligible interference between adjacent crystalline domains. Instead, FS hydrogels and AFH showed slight and strong interference between adjacent crystalline domains, respectively. The ellipsoidal SAXS pattern of AFH indicated a high orientation of nanofibrils through flow-induced alignment. When a tensile strain of 300% was applied to the AFH (AFH 300), more distinct peaks were observed in the 2D SAXS diffraction patterns, implying that the microstructures became increasingly aligned after stretching (Fig. 3i and Supplementary Fig. 12). The SAXS profiles of AFH in initial state and at 300% strain were obtained by integrating the 2D SAXS patterns (Fig. 3j). To identify the evolution of crystalline morphology in AFH during the deformation process, we investigated the average distance between adjacent crystalline domains ($D_{ac}$) and the average size of crystalline domains ($D_c$). Based on the Bragg equation ($D_{ac} = 2\pi/q_{max}$), the average distance between adjacent crystalline domains could be calculated from the scattering vector at the peak position ($q_{max}$). Besides, the average size of crystalline domains was evaluated by the correlation function analysis in SasView software. During the stretching process, the average distance between adjacent crystalline domains of AFH increased from 10.33 nm to 11.60 nm; the estimated size of crystalline domains of AFH decreased from 2.92 nm to 2.16 nm, both of which are related to the onset of microstructural changes (Fig. 3k). As schematically shown in Fig. 3l, when a tensile strain increased from 0 to 300%, the crystallographic slip and the partial unfolding of crystalline domains resulted in an increase in $D_{ac}$ and a slight decrease in $D_c$. This trend is a typical transformation of crystal morphology in semicrystalline polymers during the stretching process[43,44], which is involved in the origin of the structural toughening.

## Mechanical characterization and damage-tolerant performances of AFH

The correlation between structure and property can be demonstrated by the mechanical performance of the present materials (Fig. 4a). To verify the synergistic enhancement of flow-induced alignment and salting out, the PVA hydrogels prepared by freezing-thawing alone and freezing-soaking alone, without aligned structure and high crystallinity, were set as control samples. Both the FT hydrogels and FS hydrogels showed lower strength, toughness, and stretchability compared to AFH. In particular, the AFH with salting out for 2 hours (AFH 2 h) exhibited improved mechanical properties than the FS hydrogels with salting out for 2 h (FS 2 h), including more than 29-fold, 9-fold, and 32-fold increases in Young's modulus, tensile strength, and toughness, respectively (Fig. 4b, c). These results indicated that the unique fibrous structure of AFH facilitated the diffusion of ions in the polymer network, thereby greatly increasing the aggregation and crystallization of the polymers in a short time (Supplementary Fig. 10). In addition, tensile tests in both the longitudinal direction (FS$_∥$ and AFH$_∥$) and radial direction (FS$_⊥$ and AFH$_⊥$) were performed after salting-out treatment for 24 hours (Fig. 4d). The results manifested that AFH, with its hierarchically ordered composite structures, possessed anisotropic mechanical performance. In contrast, the mechanical properties of the isotropic FT hydrogel and FS hydrogels did not present obvious differences between the L and R directions (Supplementary Fig. 13). Meanwhile, the tensile strength of AFH in the R direction verified the strong bonding between hydrogel fibers. Notably, both the FS hydrogel (with only partial crystallinity) and the AFH 0 h (with only arrangement orientation) exhibited low mechanical properties, highlighting the critical roles of structural hierarchy and material density. With the synergy of highly oriented multi-level structure and strong crystallization, the strength and toughness of the AFH increased simultaneously (Fig. 4e and Supplementary Fig. 14). Additionally, cyclic loading with a strain of 100% was performed to observe the structural stability of the AFH. The results demonstrated that the Mullin effect existed and the change in tensile strength was negligible, further indicating that the partially damaged cross-linking points could disengage continuously to dissipate energy (Supplementary Fig. 15)[43].

Besides, the mechanical properties of AFH can be broadly and reversibly modulated by specific ions according to the Hofmeister effect, where various ions possess distinct capabilities to aggregate polymers. Two types of kosmotropic salts (ammonium sulfate and sodium citrate) were adopted as the salting-out medium to verify the tunability of the mechanical performance of AFH (Fig. 4f). After being left to salt out for 24 hours, sodium citrate (1.5 M) demonstrated a stronger salting-out effect compared to ammonium sulfate (1.5 M), which was attributed to its stronger interaction with PVA chains. The underlying mechanism is that sodium citrate has a greater ability to polarize hydrated water molecules and interfere with the hydrophobic hydration of macromolecules, which increases the structural densification and crystallinity of the hydrogel[20,41]. Thus, anisotropic fibrous hydrogel immersed in sodium citrate solution exhibited a higher tensile strength of $14 ± 1$ MPa, toughness of $154 ± 13$ MJ m$^{-3}$, and fracture energy $153 ± 8$ kJ m$^{-2}$ along the longitudinal direction (Fig. 4g and Supplementary Fig. 16). Accordingly, the various mechanical properties of AFH could be continuously regulated by changing the salting-out treatment time, ion concentration, and specific ions (Supplementary Fig. 17).

Theoretically, finite-element analysis simulation was performed to explore the fracture features of the AFH along different directions

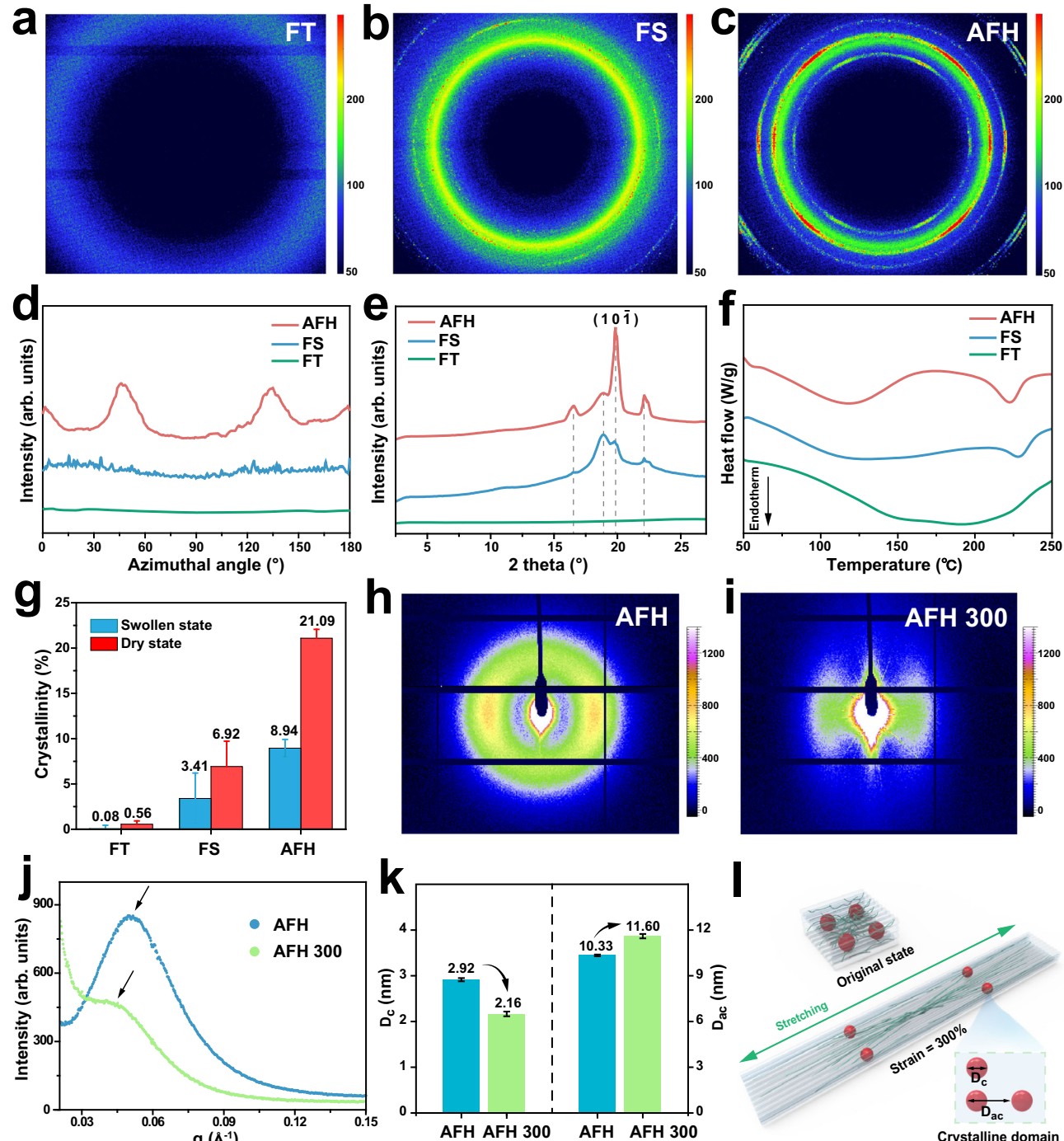

**Fig. 3 | The structural distinction of hydrogels prepared by freezing-thawing (i.e., FT), freezing-soaking (i.e., FS), and flow-induced alignment (i.e., AFH).** 2D WAXS patterns of **a** FT hydrogel, **b** FS hydrogel, and **c** AFH. **d** Scattering intensity I versus azimuthal angle θ curve of the resulting hydrogels. **e** WAXS profiles of the resulting hydrogels. **f** DSC thermographs and **g** corresponding crystallinity in the dry state and swollen state of the resulting hydrogels. **h** 2D SAXS patterns of AFH and **i** AFH 300 (when the tensile strain of AFH is 300%). **j** SAXS profiles of AFH and AFH 300. **k** The changes of the average distance between adjacent crystalline domains ($D_{ac}$) and the average size of crystalline domains ($D_c$) during stretching. **l** Schematic presentation of the changes in crystalline domains during stretching. Crystallinity, $D_{ac}$, and $D_c$ data are presented as mean values ± SD, $n = 3$ independent samples.

(Fig. 4h and Supplementary Movie 2). Before stretching, precut cracks were introduced in the L and R directions of the AFH. The inherently aligned fibrous architecture endowed AFH with a damage-tolerant characteristic to achieve crack resistance in the L direction. In contrast, the crack rapidly propagated parallel to the hydrogel fibers (R direction). Experimental and theoretical results indicated that the unique fibrous configuration effectively prevented cracks from propagating, similar to their biological counterparts. (e.g. muscle and tendon). To

illustrate the origin of the high fracture toughness in the L direction, the typical fracture behaviors of AFH were depicted schematically (Fig. 4i). When the pre-notched hydrogel was stretched, the microfibers in bridging zones underwent pulling out and fracturing to enhance the energy dissipation, while crack deflection and fiber bridging provided flaw-insensitive features to AFH. On the molecular level, the aligned nanofibrils with strong aggregation and high-density crystallization, which could serve as rigid high-functionality

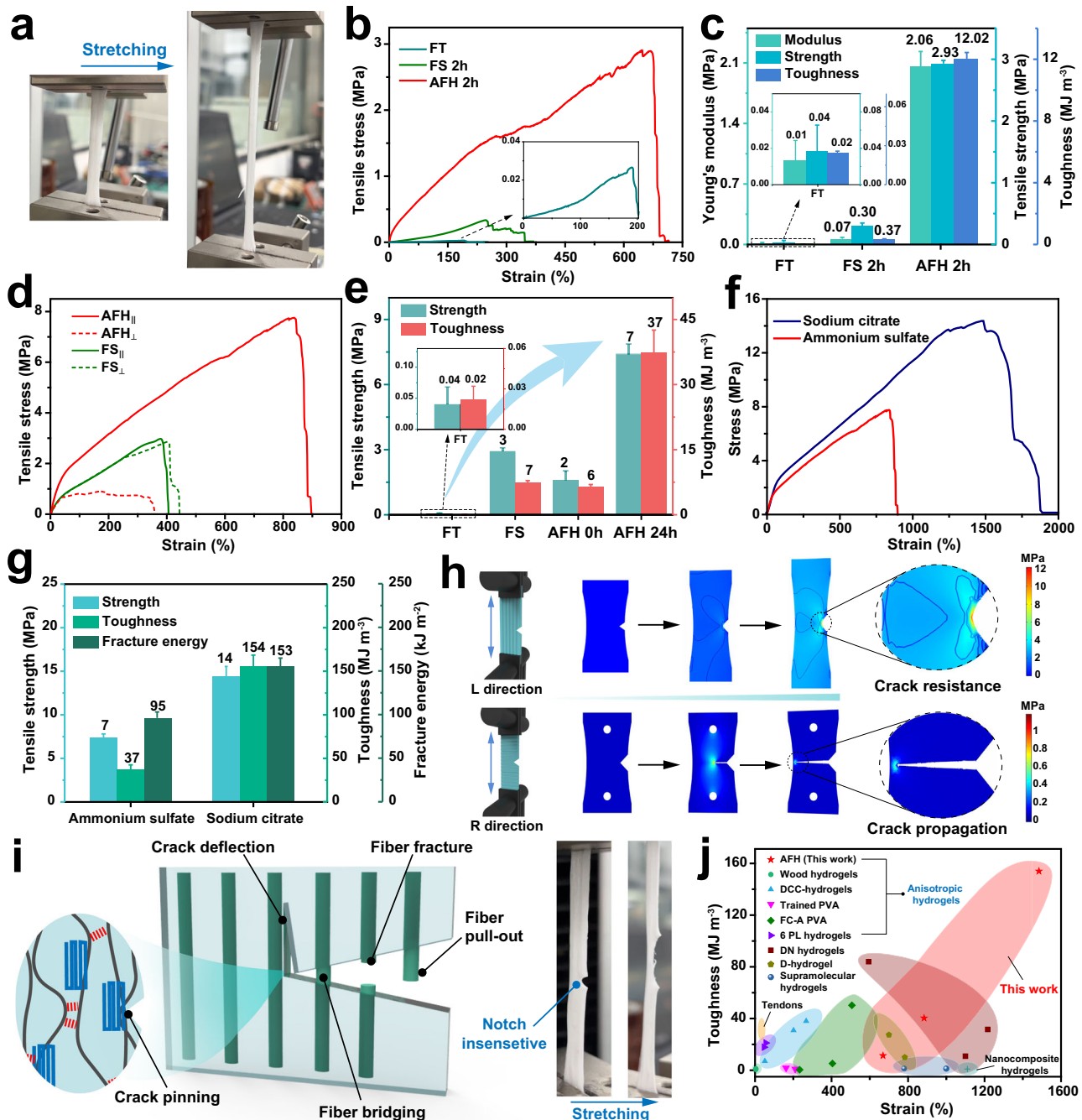

**Fig. 4 | Mechanical properties and crack insensitivity of anisotropic fibrous hydrogels. a** Optical photographs of AFH during stretching. **b** Tensile stress-strain curves of the FT hydrogel, the FS hydrogel with salting out for 2 hours (i.e., FS 2 h), and the AFH with salting out for 2 hours (i.e., AFH 2 h). **c** The average result of Young's modulus, tensile strength, and toughness of the FT hydrogels, FS hydrogels, and AFH after 2 h of salting out. **d** Tensile stress-strain curves of the FS hydrogels and the AFH hydrogels with salting out for 24 h in both the longitudinal direction ($FS_{\parallel}$ and $AFH_{\parallel}$) and radial direction ($FS_{\perp}$ and $AFH_{\perp}$). **e** Average results of tensile strength and toughness of the tested hydrogels. **f** Tensile stress-strain curves of AFH treated with 1.5 M sodium citrate and 1.5 M ammonium sulfate for 24 h in the longitudinal direction. **g** Average results of tensile strength, toughness,

and fracture energy of AFH treated with sodium citrate and ammonium sulfate. **h** Stress nephograms of AFH samples during stretching in both the L direction and R direction. **i** Anti-crack propagation mechanism of AFH hydrogels. The images on the right show AFH hydrogel with a pre-made crack. **j** Comparison chart by plotting the toughness versus strain among anisotropic hydrogels (i.e., wood hydrogel[15], hydrogels drying in confined condition (DCC-hydrogels)[23], mechanically trained PVA hydrogels[24], freeze-casting and subsequent annealing PVA hydrogels (FC-A PVA)[31], and 6-parallel laminate (6 PL) cellulose hydrogels[55]), tough hydrogels (i.e., double network (DN) hydrogels[56–58], dual-crosslinked (D) hydrogels[59,60], supramolecular hydrogels[61,62], and nanocomposite hydrogels[63]), and natural tendons[64]. Tensile data are presented as mean values ± SD, *n* = 3 independent samples.

crosslinkers, were perpendicular to the crack path, implying that the fracture would be delayed by crack pinning. Collectively, the high strength and super toughness of AFH originated from structural densification and preferentially aligned fibrous structures. As depicted in the comparison chart of Fig. 4j, the AFH synthesized in this work

exhibited excellent toughness and high stretchability, comparable to that of most anisotropic hydrogels and other tough hydrogels, even surpassing the toughness of natural tendons.

Owing to its generic applicability to various polymers, this flow-induced alignment strategy was adopted to create anisotropic

gelatin hydrogels and composite hydrogels made of PVA and gelatin. SEM images visually demonstrated the multi-scale alignment of gelatin hydrogels and composite hydrogels consisting of PVA and gelatin. (Supplementary Fig. 18). Moreover, 2D SAXS patterns of gelatin hydrogels and composite hydrogels of PVA and gelatin further revealed their highly oriented multi-level architectures (Supplementary Fig. 19). Thus, this study presented a comprehensive and adaptable method for fabricating anisotropic hydrogels that possess well-organized hierarchical structures and embedded functionalities.

## Water transport in hydrogel matrix and water purification

Rapid and significant water transport in hydrogel matrix is critical for myriad applications, including soft actuators, drug delivery, and water purification[45–47]. However, water diffusion in the hydrogel polymer network still faces many challenges owing to the poroelastic behavior[48]. As a nature-inspired hydrogel, AFH not only retained anisotropic and good mechanical performance, but it also exhibited interesting unidirectional water transport behavior through intrinsically hydrophilic microchannels (Fig. 5a). As shown in Fig. 2d–g, the unique sheath-core structure and microchannels between hydrogel fibers resembled the natural structures found in plant stems, which were responsible for the efficient transport of water, ions, and other nutrients. To assess the conduction efficiency of different hydrogel frameworks, we investigated the FT hydrogel, FS hydrogel, and AFH for water transport. During the water transport process, the AFH along the longitudinal direction (AFH$_\parallel$) demonstrated higher transport distance (14.42 mm) and speed (maximum speed at 65.75 mm s$^{-1}$) than FT hydrogel, FS hydrogel, and AFH along the radial direction (AFH$_\perp$) (Fig. 5b, c). As shown in Fig. 5d, AFH$_\parallel$ displayed fast water transport, achieving a height of 14.42 mm in 0.72 s and fully wetting the sample in 2.6 s. Moreover, AFH appeared a distinctly anisotropic transport behavior, with the transport speed of AFH$_\parallel$ being approximately 47 times faster than that of AFH$_\perp$ (Fig. 5e and Supplementary Movie 3). The results compare favorably to those of other water transport materials[47,49,50]. On the contrary, the upward distance of water in the FT and FS hydrogels with random polymer networks indicated a limited

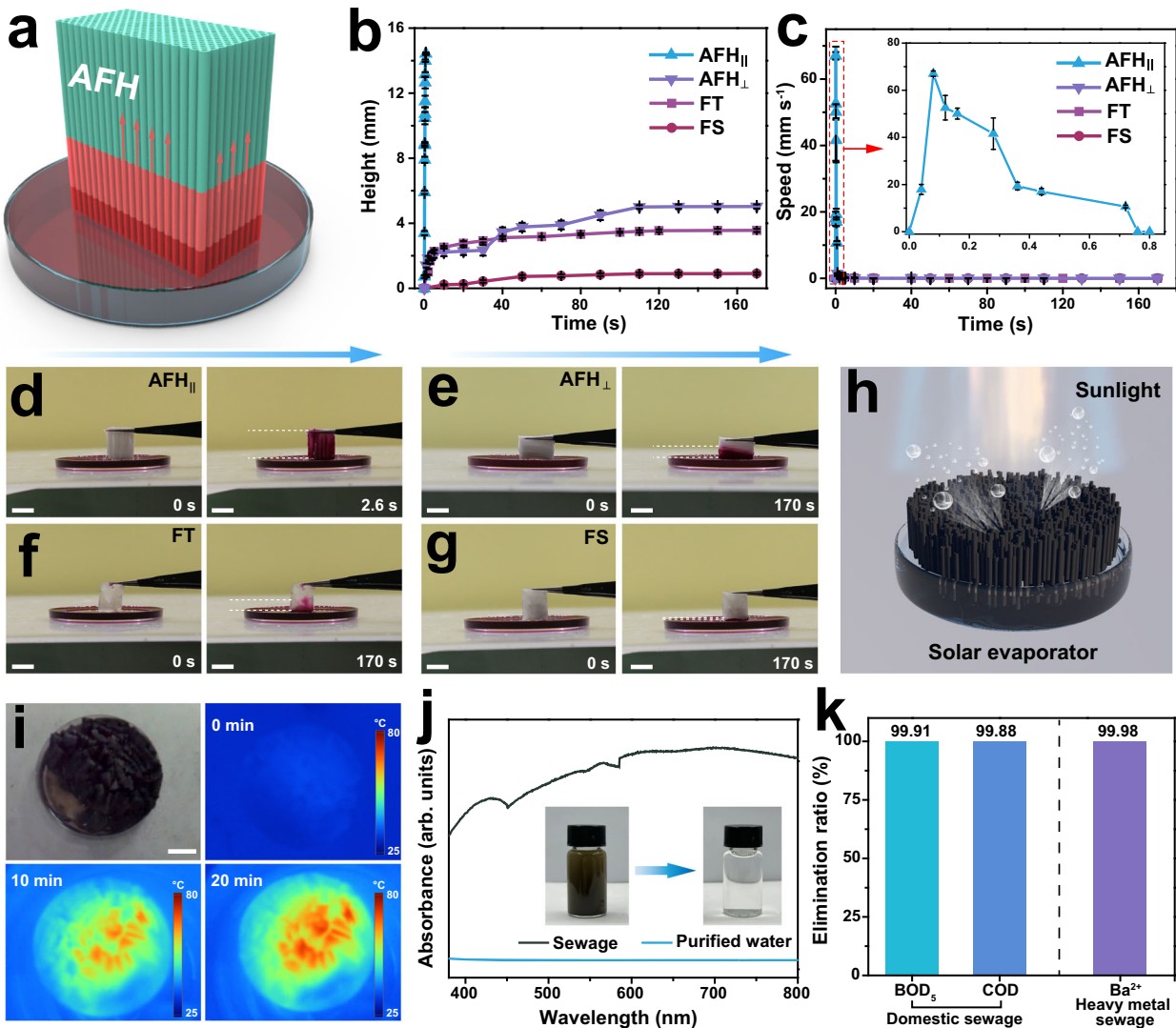

**Fig. 5 | Mass transport of the FT hydrogel, FS hydrogel, and AFH along the longitudinal and radial directions. a** Schematic illustration of AFH for mass transport in the longitudinal directions. **b** The absorbing height curves of the dye in different hydrogel structures. **c** The absorbing velocity curves of dye in different hydrogel structures. Typical digital photo of the mass transport processes of **d** AFH along the longitudinal direction (AFH$_\parallel$), **e** AFH along the radial direction (AFH$_\perp$), **f** FT hydrogel, and **g** FS hydrogel. Scale bar = 1 cm. **h** Solar-thermal hydrogel

evaporator prepared by the bioinspired structural hydrogel. **i** Facile apparatus for water purification and corresponding infrared images over time under irradiation. Scale bar = 2 cm. **j** UV-vis absorption of real sewage samples before and after purification. **k** Elimination ratios of BOD$_5$, COD, and Ba$^{2+}$ after purification by the hydrogel solar evaporator. Height and speed data are presented as mean values ± SD, n = 3 independent samples.

ability to transport water (Fig. 5f, g, and Supplementary Movie 4). The ultra-fast and anisotropic mass transport of AFH can be attributed to its aligned hierarchical structures and microchannels. These features create unidirectional capillary forces that drive the mass transport. The anisotropic mesoporous framework, integrated with the general preparation strategy, offers avenues for creating pump-free microfluidics via structural hydrogels and expands their applications in drug delivery, biological diagnosis, and water purification.

To provide a proof-of-principle verification of the concept, a solar evaporator was attempted to purify water in a low-energy way by loading poly(3,4-ethylenedioxythiophene): polystyrene sulfonate (PEDOT: PSS) on the present structural hydrogels (Fig. 5h). The AFH containing PEDOT: PSS exhibited a higher evaporation rate compared to the control group (without an evaporator) under natural sunlight. (11:30–17:30, 5 May 2023, Xiamen, China), resulting from the excellent water transport ability (Supplementary Fig. 20). Additionally, two actual sewage samples (domestic sewage and heavy metal sewage) from a local sewage treatment plant were used to assess the practical water purification capability of the hydrogel evaporator. Under the irradiation of the xenon lamp, the infrared images showed that the temperature of the sample rose promptly and reached a steady state after 20 min (Fig. 5i and Supplementary Fig. 21). The purified water was collected for ultraviolet-visible (UV-vis) absorption spectroscopy, confirming the successful removal of the majority of pollutants from the sewage (Fig. 5j). To further assess the water purification, the levels of biochemical oxygen demand in 5 days ($BOD_5$), chemical oxygen demand (COD), and barium ion were detected before and after evaporation. The results manifested that the hydrogel evaporator exerted a high-efficiency elimination, including 99.91% for $BOD_5$, 99.88% for COD, and 99.98% for barium ion, implying the potential of AFH for solar-driven clean water harvesting (Fig. 5k and details in Supplementary Table 2).

## Discussion

In summary, inspired by natural structural materials, we developed high-performance anisotropic fibrous hydrogels with a well-organized hierarchical structure using flow-induced orientation of nanofibrils. General guidance on the design and fabrication of tissue-like AFH was presented. Briefly, a precursor solution was injected into the short-distance coagulation to produce hydrogel fibers, and subsequently, the hydrogel fibers were collected and assembled directly, which firmly bound the hydrogel fibers together. After subsequent salting out treatment, the hydrated water between the polymer chains was expelled, thus resulting in a dense network. By integrating this flow-induced alignment approach with industrial wet spinning, polymer networks could be dynamically tuned without the use of reinforcements, high time/energy consumption, and cumbersome postprocessing.

Consequently, the prepared AFH showed excellent mechanical properties, for example, elongation of $1573 \pm 140\%$, strength of $14 \pm 1\,MPa$, toughness of $154 \pm 13\,MJ\,m^{-3}$, and fracture energy of $153 \pm 8\,kJ\,m^{-2}$, comparable to natural structural materials and those of other tough hydrogels. These results were attributed to the synergistic effects of structural densification and hierarchically aligned fibrous structures. By regulating the aggregation states of the polymer chains based on the Hofmeister effect, the mechanical properties of these hydrogels could be adaptable to meet specific demands. Moreover, the unique sheath-core structure and microchannels offered a promising opportunity for creating pump-free microfluidics based on anisotropic fibrous hydrogel, highlighting its potential for advanced water purification. This study elaborated multiscale strengthening and toughening mechanisms, providing insight into the design and development of soft structural matter based on simple building blocks. We envision that the presented strategy can facilitate the application of synthetic structural materials in bioengineering, drug delivery, water purification, and soft electronics.

## Methods

### Materials

Polyvinyl alcohol (PVA, M.W. 146000 - 186000, 99+% hydrolyzed, Sigma-Aldrich), gelatin (porcine skin, type A, 300 Bloom, Sigma-Aldrich), ammonium sulfate (Xilong Scientific Co.), Sodium Citrate (Sigma-Aldrich), Glutaraldehyde (50% in $H_2O$, Aladdin), hydrochloric acid (36.5–38 wt%, Sinopharm Chemical Co. Ltd.), PEDOT: PSS (1.3–1.7 wt%, Clevios PVP AI 4083, from Heraeus) and other chemicals (Aladdin) were purchased and used without further purification. Ultrapure water (18.2 MΩ; Millipore Co., USA) was used throughout the experiment.

### Fabrication of anisotropic and isotropic PVA hydrogels

A certain amount of PVA powder was vigorously stirred and heated (90 °C) for 3 h in deionized water to obtain a PVA precursor solution. After degassing by sonication for 1 hour, a homogeneous and transparent solution was obtained. Preparation of anisotropic PVA hydrogels: the PVA precursor solution (20 ml) was loaded into a syringe, and then injected into a 3 M ammonium sulfate solution (injection speed ranged from 0.6 ml min⁻¹ to 1.75 ml min⁻¹) by a syringe pump, passing through a coagulation bath less than 50 cm to obtain hydrogel fibers. The reeling bobbin (diameter: 40 mm) driven by a servo motor was adopted to collect and assemble the spun hydrogel fibers (rotation speed ranged from 15 rpm to 35 rpm). The resulting fibrous hydrogel was cut and immersed in a kosmotropic salt solution to obtain anisotropic AFH. As a comparison with isotropic PVA hydrogel, we used a PVA concentration of 10 wt%, an injection speed of 0.6 ml min⁻¹, and a rotation speed of 25 rpm. Preparation of isotropic PVA hydrogels: the PVA precursor solution was poured into Teflon molds and was frozen at −20 °C. After 8 h, the frozen samples were thawed at room temperature for 3 h to obtain FT hydrogels and immersed in a kosmotropic salt solution to obtain FS hydrogels.

### SEM characterization

To investigate the structure and surface morphology of the hydrogel, samples were frozen with liquid nitrogen and then freeze-dried. For the anisotropic hydrogel samples, after freezing with liquid nitrogen, they were cut along the directions parallel and perpendicular to the hydrogel fibers to reveal the internal structure. The lyophilized hydrogels were sputtered with gold and observed using SEM (ΣIGMA-HD, ZEISS) at an acceleration voltage of 8 kV.

### Confocal imaging

0.1 wt% fluorescein sodium salt was added to the PVA precursor solution as a fluorescent marker, followed by the synthesis of anisotropic fibrous hydrogels using the method prepared above. The 488-nm laser channel was adopted to excite fluorescence and confocal images were observed by a Leica DMi8 confocal microscope.

### AFM imaging

FT hydrogel, FS hydrogel, and AFH were lyophilized and then directly characterized by AFM in tapping mode for surface morphology. The probe tapped the surface lightly and the height of the sample surface was recorded.

### Molecular dynamics simulation of the spinning process

The PVA polymer chain was simulated by using a coarse-grained model. In the experiment, the PVA polymers with a molecular weight of 146,000 (kDa) had a degree of polymerization of about 3318. We coarse-grained the PVA chains into 100 monomers, where each CG monomer consisted of 33 repeat units. The size of the CG monomer $\sigma_0$ could be calculated by the OPLSAA all-atom force field, which took the value of about $\sigma_0 = 3$ nm. A coarse-grained solvent bead with the same diameter $\sigma = 1\sigma_0$ included 472 water molecules. The bonded interactions between neighboring monomers along chain contours of CG PVA

molecules were represented by a finite extensible nonlinear elastic (FENE) potential[51,52]:

$$U_{fene} = -0.5\kappa R_0^2 \ln\left(1 - \left(\frac{r}{R_0}\right)^2\right) + 4\varepsilon\left[\left(\frac{\sigma}{r}\right)^{12} - \left(\frac{\sigma}{r}\right)^6\right] \quad (1)$$

where $R_0 = 1.5\sigma_0$ and $\kappa = 30.0\varepsilon_0/\sigma_0^2$ were chosen for all simulations to prevent chain crossings. The nonbonded interactions between all segments were modeled as a Lennard-Jones potential.

$$U_{lj}(r) = 4\varepsilon\left[\left(\frac{\sigma}{r}\right)^{12} - \left(\frac{\sigma}{r}\right)^6 - \left(\frac{\sigma}{r_c}\right)^{12} + \left(\frac{\sigma}{r_c}\right)^6\right] \quad (2)$$

The energy strength between monomers, $\varepsilon_{mm}$, was set to 0.5 to simulate the aqueous solution environment. Here $r_c = 2.5\sigma_0$ was the cut-off distance of the interactions for all simulations. Under shear flow, the energy strengths between water-water and water-monomer were set as $\varepsilon_{ww} = 0.5$ and $\varepsilon_{wm} = 0.6$ to mimic the good solvent conditions. The open-source software LAMMPS was employed for all the molecular dynamics simulations and calculations. The end-to-end distance could be calculated by the equation:

$$R = \sqrt{\langle r^2 \rangle} \quad (3)$$

where R was the end-to-end distance, and r was the distance between the ends of a PVA chain. Further details of simulations can be found in the Supplementary Methods.

## Measurement of water content

The water content of the obtained hydrogels was tested by thermal gravimetric analysis (TGA, NETZSCH). The samples were heated from 30 °C to 150 °C at a rate of 20 °C min$^{-1}$, followed by further heating to 160 °C at a rate of 5 °C min$^{-1}$ under a nitrogen atmosphere at a flow rate of 30 mL min$^{-1}$. As a result, water content (W) can be obtained from the following equation: $W = (1 - m/m_{swollen}) \times 100\%$, where $m_{swollen}$ represents the initial mass of the hydrogel, and m stands for the residual mass of the hydrogel.

## X-ray scattering characterization

The wide-angle X-ray scattering (WAXS) measurements were performed at Xeuss 2.0 SAXS System (Xenocs, France) with an X ray of 0.154189 nm, operated at 50 kV and current 0.6 mA. The distance from the sample to the detector was 88 mm, and the exposure time was set as 300 s. The in situ small-angle X-ray scattering (SAXS) measurements were carried out at BL19U2 (Shanghai Synchrotron Radiation Facility) with an X ray of 0.1033 nm. The distance from the sample to the detector was 2721.76 mm, and the exposure time was set as 10 s. Sas-View software was adopted to fit the SAXS results to estimate the average distance between adjacent crystalline domains ($D_{ac}$) and the average size of crystalline domains ($D_c$).

## Measurement of crystallinity

The crystallinity of the resulting hydrogels was measured by a differential scanning calorimeter (DSC, NETZSCH). Excess chemical cross-linking was introduced to minimize extra crystallization in the air-drying process. The hydrogels were first soaked in the 100 mL solution consisting of 10 mL of glutaraldehyde (50% in $H_2O$) and 1 mL of hydrochloric acid (36.5–38 wt%) for 6 h, followed by soaking in deionized water for 24 h to remove excess chemicals. Subsequently, the treated samples were dried in an incubator at 37 °C for 2 h. In a typical DSC measurement, air-dried samples were first weighed and then placed in a Tzero pan. The sample was heated from 50 °C to 250 °C at a rate of 20 °C min$^{-1}$ under a nitrogen atmosphere. The results showed a broad peak at 60 °C to 180 °C, attributed to the residual water in the sample. According to the method reported previously[24,40], the mass of the residual water $m_{residual}$ can be obtained from $m_{residual} = m_{initial} \cdot H_{residual}/H^0_{water}$, where $m_{initial}$ is the mass of the sample before DSC testing, $H^0_{water} = 2260\ J\ g^{-1}$ is the latent heat of water evaporation. In addition, the curve of heat flow shows a narrow peak at 200–250 °C, corresponding to the melting of the PVA crystalline domains. Similarly, the mass of the PVA crystalline domains $m_{crystalline}$ can be obtained from $m_{crystalline} = m_{initial} \cdot H_{crystalline}/H^0_{crystalline}$, where $H^0_{crystalline} = 138.6\ J\ g^{-1}$ is the enthalpy of fusion of fully crystalline PVA measured at the equilibrium melting point[53]. Accordingly, the crystallinity in the dry state can be calculated as $X_{dry} = m_{crystalline}/(m_{initial} - m_{residual})$, and the crystallinity in the swollen state can be calculated as $X_{swollen} = X_{dry}(1 - W)$, where W represents the water content of sample measured by TGA.

## Mechanical measurements

The universal tensile machine (JHY-5000) was used to test the mechanical properties of hydrogels. The stretching speed was set as 100 mm min$^{-1}$ in the tensile test and successive loading-unloading tests. The strain was obtained as the ratio of the change in length to the initial length, and stress was obtained by dividing the force by the cross-sectional area of the sample. Young's modulus was estimated from the slope of the initial linear region of the stress-strain curve. The toughness of hydrogel was defined as the work area under the stress-strain curve up to the fracture strain. Pure shear tests were performed to calculate the fracture energy of hydrogel. Unnotched and notched samples with the same initial dimensions were stretched, where the notch size was approximately half the sample width. The fracture energy can be given by $\Gamma = U(L_c)/A$, where $L_c$ represents the critical extension in which the notch turned into a running crack, $U(L_c)$ is the work done to an unnotched sample to reach $L_c$, and A is the cross-section area of the hydrogel.

## Finite-element analysis simulation

COMSOL Multiphysics was used to simulate the fracture behaviors of AFH along different directions. Mooney-Rivlin, a classical hyperelastic model, was adopted to simulate the mechanical properties of precut AFH during stretching. To simplify the calculation, we chose Mooney-Rivlin with two parameters. By fixing a constraint on the bottom of the AFH and applying an upward load on the top of the AFH, this can be adjusted by increasing the load parameter.

## Water transport experiments

A certain amount of Congo red was added to the deionized water to increase the visibility of the water transport. The bottom of the obtained hydrogel was immersed in the dye solution, and then a digital camera was used to record the transport of the aqueous dye solution in the hydrogel.

## Water purification

PEDOT: PSS (1.3–1.7 wt%) was mixed with PVA at a mass ratio of 1:3 under stirring and heating. AFH containing PEDOT: PSS was prepared by the flow-induced alignment method, and then immersed in ammonium sulfate solution and aqueous solution sequentially to obtain a solar-driven hydrogel evaporator. The hydrogel evaporator was inserted vertically into the sewage surface and placed under natural and artificial sunlight for water purification. An artificial light source (CEAULIGHT, CEL-HXF300) was used to perform water purification of real sewage. The average light intensity at the sample surface (0.9 kW m$^{-2}$) was measured by an optical power meter (SANPOMETER, SM206). The temperature of the sample during irradiation was recorded by an infrared camera. According to previously reported methods[54], the solar-thermal conversion efficiency ($\eta_{pt}$) can

be calculated by the equation:

$$\eta_{pt} = \frac{hS(T_s - T_a) - Q_w}{\int I_{wl}(1 - 10^{-A_{wl}})} \tag{4}$$

where h stands for the heat transfer coefficient, S represents the surface area of the sample, $T_s$ is the temperature at steady state (68.2 °C), $T_a$ represents the ambient temperature (30.6 °C), $Q_w$ represents the heat consumed by water, $I_{wl}$ stands for the light intensity of a certain wavelength, and $A_{wl}$ is the corresponding absorbance. Based on Eq. (4), the solar-thermal conversion efficiency was estimated to be 77.4%. Alternatively, increasing the amount of light-absorbing filler materials could potentially enhance the solar-thermal conversion efficiency.

### Measurement of BOD$_5$, COD, and heavy metal ion
Sewage before and after purification was diluted and then sealed in culture bottles for five days. The dissolved oxygen concentration in the solution before and after cultivation was measured separately, and the oxygen consumption per liter of water can be calculated through the difference, that was, BOD$_5$. For COD measurement, the dichromate method was used to determine the COD value. Atomic absorption spectrophotometry was adopted to calculate the removal rate of barium ion.

### Data availability
Source data are provided with this paper. The data that support the findings of the current study are available in figshare with the identifier https://doi.org/10.6084/m9.figshare.24523219. Source data are provided with this paper.

### Code availability
The code that support the findings can be accessed from GitHub using the following link https://github.com/YHLinLab/End-to-end.git.

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

## Acknowledgements

This research was funded by the National Nature Science Foundation (Grant No. 12274356, 12374207), Fundamental Research Funds for the Central Universities (20720220022), and the 111 Project (B16029). We thank the staff from the BL19U2 beamline of the National Facility for Protein Science in Shanghai (NFPS) at the Shanghai Synchrotron Radiation Facility for assistance during SAXS data collection. Y.H. and Q.T. acknowledge the financial support by the open research fund of Key Laboratory of Quantum Materials and Devices (Southeast University), Ministry of Education. The super computing resources at Beijing Super Cloud Computing Center (BSCC) are acknowledged. We thank Associate Professor Lianlian Fu from Huaqiao University for her valuable assistance in the collection and analysis of SAXS data.

## Author contributions

Y.L. and P.S.L. supervised the project. S.Z. and Y.L. conceived the project. S.Z., S.W., R.S., C.F., and S.X. performed the experiments. Y.H. and Q.T. performed the coarse-grained simulations. T.F. carried out a finite-element analysis simulation. S.Z., P.S.L., and Y.L. analyzed the results and wrote the paper. All authors discussed the results and commented on the manuscript.

## Competing interests

The authors declare no competing interests.
