## [Peer Review File · Nature Communications]

Bioinspired structural hydrogels with highly ordered hierarchical orientations by flow-induced alignment of nanofibrilsREVIEWER COMMENTS

Reviewer #1 (Remarks to the Author):

In this manuscript, inspired from the natural structural materials, authors developed a novel strategy to design hierarchically structured biomimetic hydrogels by shear flow induced alignment on simple building blocks. The anisotropic structural hydrogels present ultra fast water transport. In general, it is an interesting work and the manuscript is well organized. However, there are still some issues to be addressed. A moderate revision is suggested before its acceptance.

1. More solid data should be presented in abstract section.
2. The gap areas and the new contribution in the paper should be further clarified.
3. The general introduction on the preparation, structure, properties, modifications and applications of hydrogel required further description with some recent supporting articles: Gels 9 (6), 438, 2023; e-Polymers 22, 719-732, 2022; International Journal of Biological Macromolecules 230, 123117, 2023; Polymers 14 (24), 5454, 2022; Chinese Chemical Letters 32 (10), 2923-2932, 2021; e-Polymers 22 (1), 468-477, 2022; etc.
4. Most of the contents in last paragraph of introduction can be shifted into the discussion section. This paragraph can be shortened to briefly introduce the novelty, strategy, method and important results.
5. More experimental details can be added into fig. 1.
6. The water transport properties should be compared with some previous published articles, such as: ACS nano 15 (12), 20666–20677, 2021; Cellulose 26 (11), 6653–6667, 2019; etc.
7. There are still some typos and grammar issues in the manuscript. Authors should carefully recheck the whole manuscript.
8. In order to demonstrate this journal is the right journal for your paper and also the connection of your paper to the composite community, it would be good to include some latest papers published in this journal as references.

Reviewer #2 (Remarks to the Author):

Comment:

In this work, the authors prepared anisotropic hydrogels by flow induced alignment followed by salting out in kosmotropic salt solution to induce the crystallization of the hydrogel. However, both the wet spinning method and the salting out via Hoffmeister effect are not novel in hydrogel preparation, it seems the authors did a simple combination and a detailed characterization, which however does not meet the standard of Nat. Comm. Moreover, the presentation quality is not high. Therefore, I would not recommend the publication in Nat. Comm. Here are some detailed concerns:

1. Does the shear force or extension force dominate the alignment of PVA fiber in the wet spinning. Both flow fields exist in the procedure, which one is dominate? How to prove it?
2. How is the repeatability of the average distance between adjacent crystalline domains and the

average size of crystalline domains from SAXS measurement? Due to the small detection area in SAXS measurement and the possible inhomogeneity of the sample, it is necessary to measure at different position and provide the average value and error bar.

3. How is the efficiency of the wet spinning? It is not possible for mass fabrication of aligned hydrogel by this method. Please correct the statement.

4. The comparison of mechanical properties in the longitudinal direction and radial direction is not reasonable and misleading. Aligned Fiber after coagulation would have weak fiber-fiber interaction. Exception is that one can measure the mechanical properties of a single fiber in the two different directions.

Reviewer #3 (Remarks to the Author):

This paper introduced an interesting strategy to produce anisotropic hydrogels by shear-flow-induced alignment and followed by salting out process. Similar to the wet spinning process, which shows a facile and universal approach to engineering biomimetic anisotropic hydrogels with highly aligned hierarchical fibrous structures. The aligned fibrous configuration and structure densification leads to extreme mechanical properties. And such hydrogel which was composed of the unique sheath-core structure and microchannels shows great potential for water transport and purification. However, there are still some details that the author did not fully consider. This work can be published after fully addressing the following comments.

1. In the "Preparation of bioinspired structural hydrogels" section, the author mentioned that they collected and assembled the hydrogel fibers directly after a short-distance coagulation bath, which is different from the conventional wet-spinning process, however, it is not clear that how the fibers are assembled to form the anisotropic hydrogel.

2. It is known that the injection speed and diameter of nozzle will influence the microstructure of the fibers, the author needs to indicate the parameters of the injection process and the relationship between the structure and injection parameters.

3. The authors are encouraged to explain the process in more detail. For example, Fig. 2 mentions stretch-induced orientation in illustrating the process, but that is barely discussed in the paper.

4. Sodium citrate is mentioned as a solution for more effective salting out. The authors are encouraged to explain the rationale behind using this salt.

5. The authors should explain the reason why they choose ammonium sulfate as the coagulation bath instead of other non-solvent coagulation (for example, Acetone).

6. In the “Mechanical characterization and damage-tolerant performances of AFH” section, the author mentioned that the AFH synthesized in this work exhibited excellent toughness and high stretchability that outperform existing anisotropic hydrogels and those of other tough hydrogels, however, there are some papers which are related to anisotropic hydrogel exhibit higher tensile strength and toughness than this work, for example, Hua et al. combined directional freezing and salting out to produce tough hydrogel shows an stress of 23.5 ± 2 . MPa, toughness of 210 ± 1 MJ/m³.

(<https://www.nature.com/articles/s41586-021-03212-z>), it is recommended that the author needs to modify the statement.

A point-by-point response to reviewers' comments for *Bioinspired structural hydrogels with highly ordered hierarchical orientations by shear-flow-induced alignment of nanofibrils*

REVIEWER COMMENTS

Reviewer #1 (Remarks to the Author):

In this manuscript, inspired from the natural structural materials, authors developed a novel strategy to design hierarchically structured biomimetic hydrogels by shear flow induced alignment on simple building blocks. The anisotropic structural hydrogels present ultra fast water transport. In general, it is an interesting work and the manuscript is well organized. However, there are still some issues to be addressed. A moderate revision is suggested before its acceptance.

Response: We are grateful to the reviewer for the thorough review and the highly positive comment.

1. More solid data should be presented in abstract section.

Response: We agree with this suggestion and have added solid data in the abstract section. Moreover, we have further modified the abstract to make it more logical and scientific in the revised manuscript. Detailed revision is as below.

In the revised manuscript (Page 2, line 20-33):

Herein, a simple yet versatile approach is proposed to design hierarchically structured hydrogels by flow-induced alignment of nanofibrils, without high time/energy consumption or cumbersome postprocessing. Highly aligned fibrous configuration and structural densification

are successfully achieved in anisotropic hydrogels under ambient conditions, resulting in extreme mechanical properties and damage-tolerant architectures, for example, strength of 14 ± 1 MPa, toughness of 154 ± 13 MJ m⁻³, and fracture energy of 153 ± 8 kJ m⁻². Moreover, a hydrogel mesoporous framework can implement ultra-fast and unidirectional water transport (maximum speed at 65.75 mm s⁻¹), highlighting its potential for water purification. This scalable fabrication explores a promising strategy for developing bioinspired structural hydrogels, facilitating their practical applications in biomedical and engineering fields.

2. The gap areas and the new contribution in the paper should be further clarified.

Response: Thank you and the gap areas and the new contribution in the paper have been added accordingly in the revised manuscript. Detailed revision is as below.

In the revised manuscript (Page 4, line 84-89):

This flow-induced alignment can serve as a time- and energy-saving strategy for fabricating bioinspired structural hydrogels within molecular-scale precision, elucidating its capacity for fast and large-scale production. This work represents a significant advancement in the development of hierarchically structured materials and opens up new avenues for further exploration and innovation in this captivating research area.

3. The general introduction on the preparation, structure, properties, modifications and applications of hydrogel required further description with some recent supporting articles: Gels 9 (6), 438, 2023; e-Polymers 22, 719-732, 2022; International Journal of Biological Macromolecules 230, 123117, 2023; Polymers 14 (24), 5454, 2022; Chinese Chemical Letters 32 (10), 2923-2932, 2021; e-Polymers 22 (1), 468-477, 2022; etc.

Response: Thank you for the reviewer's suggestion. The introduction section has been modified

accordingly. These recent relevant pieces of literature have been cited. The corresponding discussion has been added to the revised manuscript. Detailed revision is as below.

In the revised manuscript (Page 3, line 41-45):

Conventional synthetic hydrogels typically maintain isotropic structures with randomly oriented polymer networks, because their preparation typically involves dissolving the polymer precursor in an aqueous solution. To some extent, the presence of disordered building blocks in synthetic composites can facilitate sensing and actuation^{6,7}.

(Reference updates):

6 Li, Z. et al. Anti-freezing, recoverable and transparent conductive hydrogels co-reinforced by ethylene glycol as flexible sensors for human motion monitoring. *Int. J. Biol. Macromol.* **230**, 123117 (2023).

7 Bensefelt, T., Rothmund, P. & Lee, P. S. Ultrafast, high-strain, and strong uniaxial hydrogel actuators from recyclable nanofibril networks. *Adv. Mater.* **35**, 2300487 (2023).

4. Most of the contents in last paragraph of introduction can be shifted into the discussion section. This paragraph can be shortened to briefly introduce the novelty, strategy, method and important results.

Response: We agree with the reviewer's advice that the last paragraph of the introduction can be shortened. And we have thoroughly revised this paragraph and have added solid data to achieve accurate and scientific expression. Detailed revision is as below.

In the revised manuscript (Page 4, line 76-89):

Here, we present a novel strategy to fabricate bioinspired structural hydrogels with highly

ordered hierarchical orientations via flow-induced alignment of nanofibrils, resulting from the synergistic effect of shear and stretch in the spinning process. Due to its long-range ordered and highly dense structures, anisotropic fibrous hydrogel (AFH) exhibits remarkable mechanical properties and damage-tolerant architectures compared to its isotropic counterparts. Concretely, anisotropic fibrous hydrogels simultaneously achieve a tensile strength of up to 14 ± 1 MPa, a toughness of 154 ± 13 MJ m⁻³, and a fracture energy of 153 ± 8 kJ m⁻². Moreover, the unique microchannel structure in AFH opens up new possibilities for the development of ultra-fast and anisotropic mass transport (maximum speed at 65.75 mm s⁻¹), enabling efficient harvesting of clean water. This flow-induced alignment can serve as a time- and energy-saving strategy for fabricating bioinspired structural hydrogels within molecular-scale precision, elucidating its capacity for fast and large-scale production. This work represents a significant advancement in the development of hierarchically structured materials and opens up new avenues for further exploration and innovation in this captivating research area.

5. More experimental details can be added into fig. 1.

Response: Thanks for the reviewer's advice. We have modified Fig. 1 and added more experimental details to graphically illustrate the scheme of fluid-induced alignment of nanofibrils. Detailed revision is as below.

In the revised manuscript (Page 5, line 96-99):

The random PVA polymer chains were first subjected to flow-induced alignment and then fostered strong aggregation and crystallization by a kosmotropic salt solution, leading to an anisotropic single-composition hydrogel with a highly compact and well-aligned structure (Fig

1a).

Fig. 1 Highly aligned hierarchical fibrous hydrogels inspired by natural structural materials. a Randomly distributed PVA polymer chains transform into a highly oriented hydrogel network by flow-induced alignment.

6. The water transport properties should be compared with some previous published articles, such as: ACS nano 15 (12), 20666–20677, 2021; Cellulose 26 (11), 6653–6667, 2019; etc.

Response: Thank you and these references about water transport were added accordingly in the revised manuscript. Detailed revision is as below.

In the revised manuscript (Page 18, line 358-360; Page 18, line 374-375):

Rapid and significant water transport in hydrogel matrix is critical for myriad applications, including soft actuators, drug delivery, and water purification⁵³⁻⁵⁵.

The results compare favourably to those of other water transport materials^{55,57,58}.

(Reference updates):

55 Chen, Y. et al. Liquid transport and real-time dye purification via lotus petiole-inspired long-range-ordered anisotropic cellulose nanofibril aerogels. *ACS Nano* **15**, 20666-20677 (2021).

58 Chen, Y. et al. Anisotropic nanocellulose aerogels with ordered structures fabricated by directional freeze-drying for fast liquid transport. *Cellulose* **26**, 6653-6667 (2019).

7. There are still some typos and grammar issues in the manuscript. Authors should carefully recheck the whole manuscript.

Response: Thanks for the suggestion. We have thoroughly re-worked the writing to revise the typos and grammatical errors.

8. In order to demonstrate this journal is the right journal for your paper and also the connection of your paper to the composite community, it would be good to include some latest papers published in this journal as references.

Response: Thank you and some latest papers involved in high-performance structural hydrogels were added accordingly in the revised manuscript. Detailed revision is as below.

In the revised manuscript (Page 3, line 39-41; Page 3, line 53-54):

The design and fabrication of hydrogels have been widely explored in soft electronics, tissue engineering, and implantable devices, which is attributed to the similarity between synthetic

hydrogels and biological systems¹⁻⁵.

including electric/magnetic-field orientation¹⁴, compositing strategy¹⁵⁻¹⁸, freeze-casting¹⁹⁻²², strain alignment²³⁻²⁵, and self-assembly²⁶⁻²⁸.

(Reference updates):

3 Pan, Z. et al. Designing nanohesives for rapid, universal, and robust hydrogel adhesion.

Nat. Commun. **14**, 5378 (2023).

5 Shi, Y., Wu, B., Sun, S. & Wu, P. Aqueous spinning of robust, self-healable, and crack-

resistant hydrogel microfibers enabled by hydrogen bond nanoconfinement. *Nat. Commun.* **14**,

1370 (2023).

19 Chen, Y. et al. Multi-layered cement-hydrogel composite with high toughness, low thermal

conductivity, and self-healing capability. *Nat. Commun.* **14**, 3438 (2023).

Reviewer #2 (Remarks to the Author):

Comment:

In this work, the authors prepared anisotropic hydrogels by flow induced alignment followed by salting out in kosmotropic salt solution to induce the crystallization of the hydrogel. However, both the wet spinning method and the salting out via Hoffmeister effect are not novel in hydrogel preparation, it seems the authors did a simple combination and a detailed characterization, which however does not meet the standard of Nat. Comm. Moreover, the presentation quality is not high. Therefore, I would not recommend the publication in Nat. Comm.

Response: Thanks for the reviewer's comments. We believe that novelty is not solely limited to the invention of new technologies or materials. It also involves using existing fabrication technologies to develop new strategies for achieving the desired product. Before our work, most preparation methods for anisotropic hydrogels focused on using freeze casting and mechanical strain, which often require high time/energy consumption or cumbersome postprocessing¹⁻⁵. Our research has brought about a paradigm shift in the field. In this work, inspired by wet spinning, we presented the first example of fabricating structural hydrogels with exceptional properties via the flow-induced alignment strategy. However, unlike the traditional wet spinning strategy, which is used to produce fibers, the proposed method of flow-induced alignment can be directly applied to prepare bulk hydrogels with anisotropic structures. To underscore our work's innovation and feasibility, we have supplemented the revised manuscript with experiments and simulations, including small angle X-ray scattering using a synchrotron light source, wide angle X-ray scattering (WAXS), molecular dynamics, and spinning efficiency

validation. The merit of this work can be summarized as follows:

- a) In contrast to conventional wet spinning methods used in fiber manufacturing, the flow-induced alignment developed here offers a simple approach for fabricating hierarchical metamaterials. The traditional wet-spinning process typically involves multiple steps (such as washing and stretching) to prevent the fibers from sticking together, leading to separated fibers. Yet, the flow-induced alignment proposed in this work involves collecting and assembling the hydrogel fibers directly after a short-distance coagulation bath. This approach facilitates bonding between the as-spun fibers, resulting in a bulk hydrogel.
- b) This flow-induced alignment strategy demonstrates both generality and customizability. In addition to synthetic components (PVA), we also successfully fabricated gelatin hydrogels with well-organized hierarchical architectures using the spinning-assisted salting-out strategy. Accordingly, this method can be used to fabricate specific anisotropic hydrogels, as long as the polymer precursor is suitable for the wet spinning process. Thus, this study presents a comprehensive and adaptable method for fabricating structural hydrogels with well-ordered hierarchical architectures and embedded functionalities.
- c) Molecular-scale organization of nanofibrils can be achieved in a universal, facile, and scalable manner. In this approach, the spinning process parameters are carefully designed to regulate the alignment of nanofibrils. Various factors such as injection speed, spinneret diameter, draw ratio, polymer concentration, and coagulation bath conditions can be optimized to control the alignment of the polymer chains.
- d) Despite dramatic development in recent years, there are critical challenges in the fast and large-scale production of hierarchical hydrogel composites. Notably, owing to the novel alignment strategy that is compatible with industrial wet spinning, the production of

anisotropic fibrous hydrogels can be streamlined by incorporating this well-established industrial process with the salting-out method proposed here.

These advancements have not only enhanced the structural integrity and mechanical properties of the anisotropic hydrogels but have also opened up possibilities for their application in various fields such as biomedical engineering, drug delivery, and water purification. Our work represents a significant advancement in the development of hierarchically structured materials and opens up new avenues for further exploration and innovation in this captivating research area.

Fig. R1 Comparison between the traditional wet spinning method and flow-induced alignment strategy proposed in this work.

1. Does the shear force or extension force dominate the alignment of PVA fiber in the wet spinning.

Response: Yes. During the spinning process, the random polymer chains are forced to align with the direction of injection through the synergy of shear flow and elongational flow (Fig. R2). According to the theoretical research on wet spinning⁶⁻⁸, the shear flow controls the orientation of the polymer chains within the spinneret, where a velocity gradient perpendicular to the fluid velocity is distributed. This gradient of the velocity field creates shear forces, which tend to align the polymer chains along the direction of flow; that is shear-flow-induced alignment of nanofibrils. In addition to the shear flow, extensional flow in the coagulation bath plays a crucial role in determining the orientation and properties of polymers, which refers to a velocity gradient along the fluid velocity. During extensional flow, the polymer chains are stretched and aligned parallel to the flow direction, which results in an increase in the degree of molecular orientation in the composites; this is stretch-induced orientation. As a result, fiber shaping and orientation were dominated by both internal and external induction and alignments.

Fig. R2 The synergy of shear flow and elongational flow during the spinning process.

Both flow fields exist in the procedure, which one is dominate? How to prove it?

Response: The orientation of the polymer chain is determined by the synergistic effects of shear flow and elongational flow. To investigate the dominant factor in polymer alignment, we used molecular dynamics simulations to characterize the orientation behavior of the PVA chains (Fig. R3). We simulated the PVA polymer chain using a coarse-grained model. The shear flow and extensional flow were analyzed separately to investigate the effects of these two flow fields on the alignment of PVA chains (details in Supplementary Movie 1 and Supplementary Methods). The orientation of the PVA chain was quantified using the end-to-end distance, with the x-axis representing the injection direction or elongation direction. At low polymer concentration and shear rate, with a concentration of 10 wt% PVA aqueous solution and an injection speed of 0.6 ml/min, the PVA chains experienced minimal deformation along the x-axis due to shear flow (Fig. R3a and Fig. R3c). As the polymer concentrations and shear rates increased, the orientation of PVA chains induced by shear flow became evident (Fig. R3d and Fig. R3e). Moreover, we varied the volume ratios of PVA chains to investigate the impact of extensional flow on the orientation of PVA chains (Fig. R3b). As a result, the extensional flow in the coagulation bath was responsible for orienting the polymer, and as stretching proceeded, the polymer chains became more aligned. Based on the results of molecular dynamical simulations and spinning parameters in this work, it has been found that the orientation of the PVA chains during the spinning process is primarily influenced by the extensional flow (Fig. R3f-h).

We conclude that between the aforementioned mechanisms of the shear flow and elongational flow, which both have important roles during the spinning process, the effect of the elongational flow was particularly prominent in this work for simultaneous high orientation and extreme mechanical properties. Accordingly, the title has been adjusted to “Bioinspired

structural hydrogels with highly ordered hierarchical orientations by flow-induced alignment of nanofibrils” which would accurately express the strategy proposed in this work.

Fig. R3 Molecular dynamics simulations of PVA chains in **a** the injection process (dominated by shear flow) and **b** the stretching process (dominated by elongational flow). **c-e** The Cartesian coordinate components of the end-to-end distance of PVA chains under shear flow at polymer concentration from 10 wt% to 25 wt%, and injection speed from 0.6ml/min to 3.5 ml/min. **f-h** The Cartesian coordinate components of the end-to-end distance of PVA chains under extensional flow at various volume ratios.

2. How is the repeatability of the average distance between adjacent crystalline domains and the average size of crystalline domains from SAXS measurement? Due to the small detection area in SAXS measurement and the possible inhomogeneity of the sample, it is necessary to measure at different position and provide the average value and error bar.

Response: Thanks for the reviewer's suggestion. To obtain higher-quality data, we conduct SAXS measurements on at least three samples per experimental group using synchrotron radiation (BL19U2 beamline at the Shanghai Synchrotron Radiation Facility). Due to the high intensity and resolution of synchrotron radiation, we can obtain clear and accurate SAXS patterns, which in turn enable us to calculate more precise results. Accordingly, the repeatability of the average distance between adjacent crystalline domains (D_{ac}) and the average size of crystalline domains (D_c) from SAXS measurement are repeatable. This result demonstrates that the resulting structural hydrogel is homogeneous, and that this fabrication method can be replicated. Detailed revision is as below.

In the revised manuscript (Page 12, line 251-258; Page 13, line 259-268):

During the stretching process, the average distance between adjacent crystalline domains of AFH increased from 10.33 nm to 11.60 nm; the estimated size of crystalline domains of AFH decreased from 2.92 nm to 2.16 nm, both of which are related to the onset of microstructural changes (Fig. 3k). As schematically shown in Fig. 3l, when a tensile strain increased from 0 to 300%, the crystallographic slip and the partial unfolding of crystalline domains resulted in an increase in D_{ac} and a slight decrease in D_c . This trend is a typical transformation of crystal morphology in semicrystalline polymers during the stretching process^{43,44}, which is involved in the origin of the structural toughening.

Fig. 3 **h** 2D SAXS patterns of AFH. **i** 2D SAXS patterns when the tensile strain of AFH is 300% (AFH 300). **j** SAXS profiles of AFH were obtained under strains at 0% and 300%. **k** The changes of the average distance between adjacent crystalline domains (D_{ac}) and the average size of crystalline domains (D_a) in AFH during stretching. **l** Schematic presentation of the changes in AFH crystalline domains during stretching.

3. How is the efficiency of the wet spinning? It is not possible for mass fabrication of aligned hydrogel by this method. Please correct the statement.

Response: In general, the efficiency of wet spinning depends on the injection speed, nozzle diameter, and the number of nozzles. In this work, we have designed a straightforward, versatile, and scalable spinning approach to fabricate bioinspired structural hydrogel. Using a spinneret with a single nozzle as an example, we set the injection speed to 0.6ml/min and the nozzle diameter to 0.65mm, which allows the as-spun fibers to maintain a good orientation. With these parameters, it takes approximately half an hour to complete the spinning process for a 20ml solution of 10 wt% PVA precursors. As the number of nozzles increases, the time required for spinning gradually decreases (Table R1 and Fig. R4), indicating that when there are more

nozzles in use, the spinning process takes less time to complete.

Fig. R4 Effect of different nozzles number on spinning efficiency. **a-d** The number of nozzles is gradually increased from one to four. Scale bar = 1 cm. **e** Anisotropic hydrogel obtained by spinning 20ml PVA precursors. Scale bar = 1 cm. **f** Spinnerets with different nozzles number. Scale bar = 1 cm. **g** Spinneret with a large number of nozzles for industrial wet spinning.

This result suggests that increasing the number of nozzles can be an effective way to optimize and streamline the spinning process, resulting in greater efficiency and productivity. We utilized a limited number of nozzles to demonstrate the applicability of wet spinning in preparing anisotropic hydrogels. In fact, industrial wet spinning usually uses spinnerets with

dozens to thousands of nozzles, which can be customized according to the specific practical application, making large-scale production of anisotropic fibrous hydrogels feasible (Fig R4g).

Table R1. The efficiency of the flow-induced alignment of nanofibrils.

Number of nozzles	Injection volume (ml)	Injection speed (ml/min)	Nozzle diameter (mm)	Spinning time (min)
1	20	0.6	0.65	33.33
2	20	1.2	0.65	16.66
3	20	1.8	0.65	11.11
4	20	2.4	0.65	8.33

The corresponding discussion has been added to the revised manuscript. Detailed revision is as below.

In the revised manuscript (Page 9, line 190-201):

Since wet spinning technology has been highly industrialized, this flow-induced alignment strategy is expected to enable the mass production of structural materials. Spinnerets with varying numbers of nozzles were utilized to perform the spinning process and assess the potential for large-scale production of biomimetic hydrogels (Supplementary Fig. 9). A nozzle with a 0.65 mm inner diameter took approximately 30 minutes to complete the spinning of 20 ml PVA aqueous solution at an injection speed of 0.6 ml/min. As the number of nozzles and injection speed increased, the efficiency of AFH preparation significantly improved. (details in Supplementary Table 1). In this work, we utilized a limited number of nozzles to demonstrate the applicability of wet spinning in preparing anisotropic hydrogels. In fact, industrial wet spinning usually uses spinnerets with thousands of nozzles, which can be customized according to the specific practical application, making large-scale production of anisotropic fibrous hydrogels feasible.

4. The comparison of mechanical properties in the longitudinal direction and radial direction is not reasonable and misleading. Aligned Fiber after coagulation would have weak fiber-fiber interaction. Exception is that one can measure the mechanical properties of a single fiber in the two different directions.

Response: Thanks for the reviewer's comments. The anisotropy of the bulk hydrogels can be demonstrated by comparing the tensile strength in the longitudinal and radial directions. Yes, the interaction between the fibers weakens during the coagulation. The anisotropic mechanical properties of AFH can be primarily attributed to this key factor. However, we intend to create a hierarchically oriented structural hydrogel, which is a strong bulk material instead of fibrous materials. Testing the mechanical properties of a single fiber in the radial direction is therefore redundant. Nonetheless, testing the mechanical properties of a single fiber in the longitudinal direction can be used to optimize spinning parameters. We further investigated the tensile strength of individual fibers. The results showed that single hydrogel fiber simultaneously achieved tensile strength up to 16 ± 1 MPa, tensile strain of $1924 \pm 150\%$, and toughness of 169 ± 14 MJ m⁻³ (Fig. R5).

Fig. R5 Mechanical properties test of single hydrogel fiber. **a** Tensile stress-strain curves of single hydrogel fiber with salting out for 24 hours. **b** Average results of strain, tensile stress, and toughness of the tested hydrogel fiber.

Reviewer #3 (Remarks to the Author):

This paper introduced an interesting strategy to produce anisotropic hydrogels by shear-flow-induced alignment and followed by salting out process. Similar to the wet spinning process, which shows a facile and universal approach to engineering biomimetic anisotropic hydrogels with highly aligned hierarchical fibrous structures. The aligned fibrous configuration and structure densification leads to extreme mechanical properties. And such hydrogel which was composed of the unique sheath-core structure and microchannels shows great potential for water transport and purification. However, there are still some details that the author did not fully consider. This work can be published after fully addressing the following comments.

Response: We sincerely thank the reviewer for the thoughtful review and for all these constructive comments toward improving our manuscript. We have revised the manuscript based on the reviewer's suggestions. The point-by-point responses are provided below:

1. In the “Preparation of bioinspired structural hydrogels” section, the author mentioned that they collected and assembled the hydrogel fibers directly after a short-distance coagulation bath, which is different from the conventional wet-spinning process, however, it is not clear that how the fibers are assembled to form the anisotropic hydrogel.

Response: Thanks for the reviewer’s comments and the process of forming anisotropic hydrogels through fibers has been described in detail. As shown in Fig. R6, the traditional wet-spinning process typically involves multiple steps (such as washing and stretching) to prevent the fibers from sticking together, leading to separated fibers. Yet, the simplified spinning process proposed in this work involves collecting and assembling the hydrogel fibers directly

after a short-distance coagulation bath. This was attributed to the fact that the short-distance coagulation has not fully shaped the hydrogel fibers through the salting-out effect, leading to the assembly of adjacent as-spun fibers, which can be verified by microscopic morphology and mechanical characterization. In this case, our strategy for spinning fibers is more efficient than traditional wet spinning techniques since the spinning process undergoes only one step.

Fig. R6 Comparison between the traditional wet spinning method and flow-induced alignment strategy proposed in this work.

The corresponding discussion has been added to the revised manuscript. Detailed revision is as below.

In the revised manuscript (Page 8, line 160-171):

Notably, the conventional wet-spinning process usually includes several post-cleaning steps to prevent the fibers from sticking together, whereas the simplified steps we proposed here collect and assemble the hydrogel fibers directly after a short-distance coagulation bath. The reeling bobbin driven by a servo motor was employed to tightly assemble the spun hydrogel fibers, where a certain number of hydrogen bonds might form between adjacent nascent fibers (Fig. 2b). This was attributed to the fact that the short-distance coagulation may not have fully shaped the hydrogel fibers through the salting-out effect, resulting in the assembly of adjacent as-spun fibers, which can be verified by microscopic morphology and mechanical characterization. Subsequently, the resultant hydrogels with highly aligned fibrous structures were cut and immersed in a kosmotropic salt solution to achieve structural densification via hydrogen bonds and crystalline domains (Supplementary Fig. 7).

Fig. 2 Design strategy for fabricating bioinspired structural hydrogels. e Schematic diagram of the bonding between anisotropic hydrogel fibers.

2. It is known that the injection speed and diameter of nozzle will influence the microstructure of the fibers, the author needs to indicate the parameters of the injection process and the relationship between the structure and injection parameters.

Response: Thank you for the very insightful points. We investigated the impact of injection parameters, specifically the draw ratio (ratio of collection speed to injection speed) and diameter of nozzle, on the structure of the hydrogel fiber. The results indicated that as the draw ratio increases and the nozzle diameter decreases, the fiber diameter gradually decreases, as depicted in the SEM images of Supplementary Fig. 5. Within this framework, spinning parameters can be optimized to control the alignment of the polymer chains, allowing the preparation of structural hydrogels to be customized. The corresponding discussion has been added to the revised manuscript. Detailed revision is as below.

In the revised manuscript (Page 7, line 157-160):

Supplementary Figure 5. Effect of drawing ratio and nozzle diameter on hydrogel fibers

during the spinning process. SEM images of hydrogel fibers obtained with various (a-c) draw ratios and (d-e) nozzle diameters. Average results for fiber diameter at different (g) draw ratios and (h) nozzle diameters.

To study the impact of spinning parameters on the structure of hydrogel fibers, we employed various spinneret diameters and draw ratios to achieve flow-induced alignment. As depicted in the SEM images (Supplementary Fig. 5), the fiber diameter gradually decreases with increasing draw ratio and decreasing nozzle diameter.

3. The authors are encouraged to explain the process in more detail. For example, Fig. 2 mentions stretch-induced orientation in illustrating the process, but that is barely discussed in the paper.

Response: Thank you for the comments and the detail of flow-induced alignment in the spinning process is addressed. The synergistic effects of shear flow and elongational flow in the spinning process dominate the orientation of PVA chains. Specifically, the shear flow in the spinneret controls the alignment of the polymer chains, while extensional flow in the coagulation further enhances the orientation of the polymer chains along the injection direction. Moreover, the synergy of shear flow and elongational flow in the spinning process can be demonstrated by the computer simulations. Based on the results of molecular dynamical simulations and the spinning parameters in this work, it has been found that the orientation of the PVA chains during the spinning process is primarily influenced by the extensional flow (Supplementary Fig. 4).

The corresponding discussion has been added to the revised manuscript. Detailed revision is as below.

In the revised manuscript (Page 7, line 129-156):

According to the theoretical research on wet spinning^{33,35,36}, the shear flow controls the

orientation of the polymer chains within the spinneret, where a velocity gradient perpendicular to the fluid velocity is distributed. This gradient of the velocity field creates shear forces, which tend to align the polymer chains along the direction of flow; that is shear-flow-induced alignment of nanofibrils. In addition to the shear flow, extensional flow in the coagulation bath plays a crucial role in determining the orientation and properties of polymers, which refers to a velocity gradient along the fluid velocity. During extensional flow, the polymer chains are stretched and aligned parallel to the flow direction, which results in an increase in the degree of molecular orientation in the composites; this is stretch-induced orientation. As a result, fiber shaping and orientation were dominated by both internal and external induction and alignments. To illustrate the influence of these two flow fields on the orientation of PVA chains in this work, computer simulations were performed. We simulated the PVA polymer chain using a coarse-grained model. The orientation of the PVA chain was quantified using the end-to-end distance, with the x-axis representing the shear or elongation direction. The shear flow and extensional flow were analyzed separately to investigate the effects of these two flow fields on the alignment of PVA chains (details in Supplementary Movie 1 and Supplementary Methods). At low polymer concentration and shear rate, with a concentration of 10 wt% PVA aqueous solution and an injection speed of 0.6 ml/min, the PVA chains experienced minimal deformation along the x-axis due to shear flow. As the polymer concentrations and shear rates increased, reaching a concentration of 25 wt% for the PVA aqueous solution and an injection speed of 1.75 ml/min, the orientation of PVA chains induced by shear flow became evident (Fig. 2b and Fig. 2d). Moreover, we varied the volume ratios of PVA chains to investigate the impact of extensional flow on the orientation of PVA chains. Accordingly, the extensional flow in the coagulation bath was responsible for orienting the polymer, and as stretching proceeded, the

polymer chains became more aligned (Fig. 2c and Fig. 2e). Based on the results of molecular dynamical simulations and the spinning parameters in this work, it has been found that the orientation of the PVA chains during the spinning process is primarily influenced by the extensional flow (Supplementary Fig. 4).

Fig. 2 Design strategy for fabricating bioinspired structural hydrogels. **a** Fabrication procedure for hydrogel polymer networks using the flow-induced alignment. Molecular dynamical simulations of PVA chains in **b** the injection process and **c** stretching process. **d** The Cartesian coordinate components of the end-

to-end distance of PVA chains under two shear flow conditions: at low polymer concentration and shear rate, and high polymer concentration and shear rate. **e** The Cartesian coordinate components of the end-to-end distance of PVA chains under extensional flow at various volume ratios. **f** Schematic diagram of the bonding between hydrogel fibers. **g** Typical photographs of anisotropic fibrous hydrogels with longitudinal direction (L direction) and radial direction (R direction). Scale bar = 1 cm. **h** SEM cross-sectional image of anisotropic fibrous hydrogels in the R direction. Scale bar = 300 μm . **i** Magnified SEM image of (h) demonstrating the detailed microstructure. Scale bar = 40 μm . **j** SEM cross-sectional image of anisotropic fibrous hydrogels in the L direction. Scale bar = 100 μm . **k** Magnified SEM image of (j) demonstrating the detailed microstructure. Scale bar = 20 μm .

4. Sodium citrate is mentioned as a solution for more effective salting out. The authors are encouraged to explain the rationale behind using this salt.

Response: Thanks for the reviewer's precious suggestion. According to the Hofmeister effect, also known as the Ion-specific effect, different ions have distinct abilities to precipitate polymers. It has been reported that various ions have a specific effect on the gelation of PVA. Ion-specific gelation arises from the various interaction modes with PVA polymer chains, which can lead to either salting-out or salting-in effects. For example, PVA can precipitate strongly to form gels in the presence of kosmotropic ions (such as citrate³⁻, SO₄²⁻, CO₃²⁻), but it dissolves quite well in certain ions (such as Mg²⁺, Cl⁻, NO₃⁻). The underlying mechanism is that kosmotropic ions can facilitate the removal of water molecules between PVA chains and enhance the self-aggregation of PVA chains. On the other hand, certain ions can introduce additional charges to the PVA chains, thereby increasing their solubility. Since sodium citrate has a greater ability to polarize hydrated water molecules and interfere with the hydrophobic hydration of macromolecules, AFH immersed in sodium citrate solution exhibited higher mechanical properties. The corresponding discussion has been added to the revised manuscript.

Detailed revision is as below.

In the revised manuscript (Page 15, line 301-308):

After being left to salt out for 24 hours, sodium citrate (1.5 M) demonstrated a stronger salting-out effect compared to ammonium sulfate (1.5 M), which was attributed to its stronger interaction with PVA chains. The underlying mechanism is that sodium citrate has a greater ability to polarize hydrated water molecules and interfere with the hydrophobic hydration of macromolecules, which increases the structural densification and crystallinity of the hydrogel^{20,41}. Thus, anisotropic fibrous hydrogel immersed in sodium citrate solution exhibited a higher tensile strength of 14 ± 1 MPa, toughness of 154 ± 13 MJ m⁻³, and fracture energy 153 ± 8 kJ m⁻² along the longitudinal direction (Fig. 4g and Supplementary Fig. 16).

5. The authors should explain the reason why they choose ammonium sulfate as the coagulation bath instead of other non-solvent coagulation (for example, Acetone).

Response: As suggested by the reviewer, we have added the reason for choosing ammonium sulfate solution as the coagulation bath in the revised manuscript. Compared to other non-solvent coagulants (such as Acetone), ammonium sulfate offers several advantages, including non-volatility, high water solubility, and low cost. These advantages allow for the implementation of the flow-induced alignment strategy under mild conditions, making the industrialization process environmentally friendly and cost-effective. In addition, ammonium sulfate is commonly used in the coagulation bath of industrial wet spinning processes because of its high yield and availability. The corresponding discussion has been added to the revised manuscript. Detailed revision is as below.

In the revised manuscript (Page 4, line 94-99):

We chose ammonium sulfate solution as the coagulation since it has low volatility, high water solubility, and is cost-effective and environmentally friendly compared to other non-solvent coagulants. The random PVA polymer chains were first subjected to flow-induced alignment and then fostered strong aggregation and crystallization by a kosmotropic salt solution, leading to an anisotropic single-composition hydrogel with a highly compact and well-aligned structure (Fig 1a).

6. In the “Mechanical characterization and damage-tolerant performances of AFH” section, the author mentioned that the AFH synthesized in this work exhibited excellent toughness and high stretchability that outperform existing anisotropic hydrogels and those of other tough hydrogels, however, there are some papers which are related to anisotropic hydrogel exhibit higher tensile strength and toughness than this work, for example, Hua et al. combined directional freezing and salting out to produce tough hydrogel shows an stress of 23.5 ± 2 . MPa, toughness of 210 ± 1 MJ/m⁻³.

(<https://www.nature.com/articles/s41586-021-03212-z>), it is recommended that the author needs to modify the statement.

Response: Thank you for the suggestion and the statement related to the mechanical properties of anisotropic hydrogels is modified. The corresponding discussion has been added in the revised manuscript and this reference has been cited. Detailed revision is as below.

In the revised manuscript (Page 16, line 327-330):

As depicted in the comparison chart of Fig. 4j, the AFH synthesized in this work exhibited excellent toughness and high stretchability, comparable to that of most anisotropic hydrogels and other tough hydrogels, even surpassing the toughness of natural tendons.

References

- 1 Fan, H. & Gong, J. P. Fabrication of bioinspired hydrogels: challenges and opportunities. *Macromolecules* **53**, 2769-2782 (2020).
- 2 Hua, M. *et al.* Strong tough hydrogels via the synergy of freeze-casting and salting out. *Nature* **590**, 594-599 (2021).
- 3 Lin, S., Liu, J., Liu, X. & Zhao, X. Muscle-like fatigue-resistant hydrogels by mechanical training. *Proc. Natl Acad. Sci. USA* **116**, 10244-10249 (2019).
- 4 Mredha, M. T. I. & Jeon, I. Biomimetic anisotropic hydrogels: Advanced fabrication strategies, extraordinary functionalities, and broad applications. *Prog. Mater. Sci.* **124**, 100870 (2022).
- 5 Zhao, Z., Fang, R., Rong, Q. & Liu, M. Bioinspired nanocomposite hydrogels with highly ordered structures. *Adv. Mater.* **29**, 1703045 (2017).
- 6 Rohani Shirvan, A., Nouri, A. & Sutti, A. A perspective on the wet spinning process and its advancements in biomedical sciences. *Eur. Polym. J.* **181**, 111681 (2022).
- 7 Vigolo, B. *et al.* Macroscopic fibers and ribbons of oriented carbon nanotubes. *Science* **290**, 1331-1334 (2000).
- 8 Fang, B., Chang, D., Xu, Z. & Gao, C. A review on graphene fibers: expectations, advances, and prospects. *Adv. Mater.* **32**, e1902664 (2020).

REVIEWERS' COMMENTS

Reviewer #1 (Remarks to the Author):

Authors have addressed all the comments well. An acceptance is suggested.

Reviewer #2 (Remarks to the Author):

No further comments

Reviewer #3 (Remarks to the Author):

The authors have addressed all the comments in detail. This article is recommended for acceptance.